# Global mapping of GalNAc-T isoform-specificities and O-glycosylation site-occupancy in a tissue-forming human cell line

Mathias I. Nielsen [1,5], Noortje de Haan[1,5], Weston Kightlinger[1,2], Zilu Ye [1,3], Sally Dabelsteen[4], Minyan Li[1], Michael C. Jewett [2], Ieva Bagdonaite [1], Sergey Y. Vakhrushev [1] ✉ & Hans H. Wandall [1] ✉

Mucin-type-O-glycosylation on proteins is integrally involved in human health and disease and is coordinated by an enzyme family of 20 *N*-acetyl-galactosaminyltransferases (GalNAc-Ts). Detailed knowledge on the biological effects of site-specific O-glycosylation is limited due to lack of information on specific glycosylation enzyme activities and O-glycosylation site-occupancies. Here we present a systematic analysis of the isoform-specific targets of all GalNAc-Ts expressed within a tissue-forming human skin cell line, and demonstrate biologically significant effects of O-glycan initiation on epithelial formation. We find over 300 unique glycosylation sites across a diverse set of proteins specifically regulated by one of the GalNAc-T isoforms, consistent with their impact on the tissue phenotypes. Notably, we discover a high variability in the O-glycosylation site-occupancy of 70 glycosylated regions of secreted proteins. These findings revisit the relevance of individual O-glycosylation sites in the proteome, and provide an approach to establish which sites drive biological functions.

Glycosylation expands the structural diversity of the proteome, affecting protein folding, stability, processing, trafficking, immune recognition, and biological activity[1]. Several types of protein-glycosylation exist, including *N*-linked glycosylation, C-mannosylation, and different forms of O-glycosylation, which at large comprise of O-GalNAc-glycosylation, O-mannosylation, O-fucosylation, O-glucosylation, and the addition of glycosaminoglycans to O-xylose[2]. O-GalNAc-glycosylation is among the most abundant and diverse types of O-glycosylation and initiated by 20 differentially expressed polypeptide GalNAc-transferases (GalNAc-Ts) with overlapping but distinct substrate specificities[3,4]. These enzymes offer the possibility for dynamic and specific fine-tuning of defined protein functions. The biological significance of site-specific O-GalNAc-glycosylation

(hereafter O-glycosylation) in human biology is illustrated in animal models, human genome-wide association studies, and in patients where deleterious mutations in individual GalNAc-Ts are linked to specific phenotypes and human diseases[3]. These include immune disorders, metabolic disorders, dyslipidemia, bone disorders and calcium-homeostasis imbalance, neurodegenerative diseases, kidney disease, and many different aspects of cancer biology[3]. To better devise new treatment modalities for such diseases there is an intense and continued interest in better understanding how O-glycans impact human biology and pathology.

The initiation of O-glycans takes place in the Golgi apparatus and involves the transfer of *N*-acetylgalactosamine (GalNAc) from a uridine diphosphate-GalNAc (UDP-GalNAc) to a Ser or Thr (and less commonly

[1]Copenhagen Center for Glycomics, Departments of Cellular and Molecular Medicine, Faculty of Health and Medical Sciences, University of Copenhagen, Copenhagen, Denmark. [2]Department of Chemical and Biological Engineering and Center for Synthetic Biology, Northwestern University, Evanston, IL 60208, USA. [3]Novo Nordisk Foundation Center for Protein Research, Faculty of Health and Medical Sciences, University of Copenhagen, Copenhagen, Denmark. [4]Department of Oral Medicine and Pathology, School of Dentistry, University of Copenhagen, Copenhagen, Denmark. [5]These authors contributed equally: Mathias I. Nielsen, Noortje de Haan. ✉e-mail: seva@sund.ku.dk; hhw@sund.ku.dk

Tyr) residue within proteins, catalyzed by one of the 20 GalNAc-Ts[3,4]. GalNAc-Ts contain a ricin-type lectin domain which recognizes GalNAc-residues 6–17 amino-acids away from the acceptor site, and thereby control and accelerate follow-up glycosylation events[5]. The control of O-glycosylation by prior addition of GalNAc residues, and their further elaboration into several core structures, has made it difficult to determine the individual glycosylation sites governed by each GalNAc-T, and hence to understand how these 20 enzyme isoforms collectively regulate the glycoproteomes of cells.

A first step in defining the biological significance of O-glycosylation is to map the position of O-glycosylation sites in native proteins. During the last decade, the development of sophisticated mass spectrometry methods, in combination with genetic engineering, metabolic labeling, and various enrichment techniques, has allowed a systematic analysis of the O-glycoproteome in different cell lines, and a few cases of native tissues[6–9]. The analyses have demonstrated that O-glycans represent one of the most abundant and diverse types of glycans, with more than 10,000 sites found on ~80% of the proteins traveling through the secretory pathway[6,7]. Moreover, mapping the location of the O-glycans to protein domains with known functions has made it obvious that O-glycosylation sites are located in close proximity to many proteolytic cleavage sites and help fine-tune proprotein processing and ectodomain shedding[10]. In addition, O-glycans are found in specialized regions, such as mucin domains and the class A repeats of the low-density lipoprotein receptor (LDLR) and different LDLR-related proteins (LRPs), to provide structural stability[3,11]. Collectively, these findings help to explain the link between defects in GalNAc-Ts and certain disease phenotypes, such as effects on calcium metabolism as seen in familial tumoral calcinosis[12,13], or the lipid dysregulation associated with GalNAc-T2[14–17].

Despite this progress, we still only understand a fraction of the potential functions related to O-glycans at specific sites. Hence, there is a need for more effective ways to identify the specific O-glycans on select proteins that drive protein functions. Among the possible solutions is to use defined model systems that make it possible to link individual O-glycosylation sites governed by each GalNAc-T to possible biological roles. In addition, understanding the occupancy of the individual O-glycosylation sites would help us identify which sites are most likely to drive specific functions. Unfortunately, this information is inherently lost with the current enrichment strategies for O-glycoproteomic analyses[18].

Here, we combine genetic engineering and differential glycoproteomics to perform a systematic precision mapping of the site-specificity of nine GalNAc-Ts (T1, T2, T3, T6, T7, T10, T11, T14, and T18) expressed within the tissue-forming human cell line N/TERT-1 (Fig. 1a). We demonstrated that while the majority of the glycoproteome was not affected by the loss of individual GalNAc-Ts, distinct subsets of glycosites were regulated exclusively by a single isoform, providing cells with the ability to dynamically control the glycoproteome. Additionally, we show that it is possible to evaluate the site occupancy of the identified glycosylation sites using the non-enriched secreted material of wild type and *COSMC*⁻/⁻ N/TERT-1 cells (Fig. 1b), which helps to determine important sites for future functional evaluation. Together, these data not only present a comprehensive, consistent, and improved map of regulated O-glycosites in native-like epithelial cells, but also support the selection of promising leads for uncovering novel functions of site-specific O-GalNAc glycans in tissue differentiation and homeostasis.

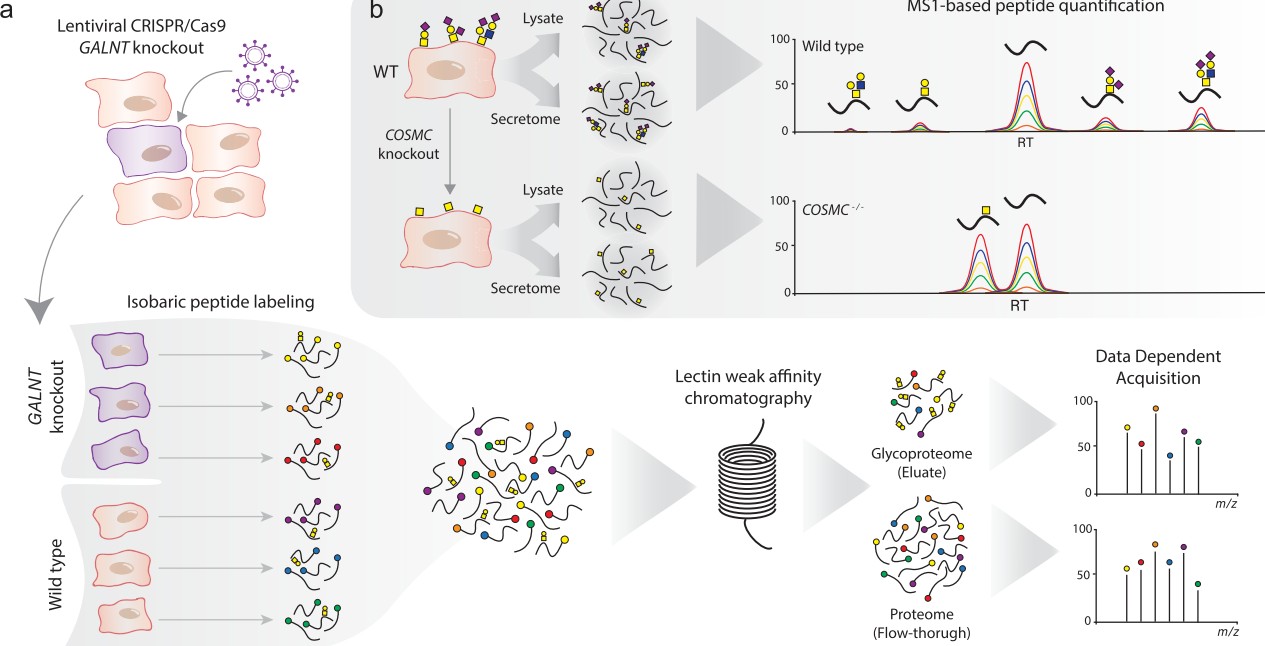

**Fig. 1 | Scheme of the applied workflow. a** *GALNT* knockouts in N/TERT-1 cell lines were generated using lentiviral transduction to deliver the CRISPR/Cas9 machinery. Total cell lysates were collected from three technical wild-type replicates and three biological replicate knockout clones, and extracted proteins were reduced, alkylated, and digested with trypsin. Purified peptides were labeled using tandem mass tags (TMT), mixed, and treated with neuraminidase prior to jacalin LWAC enrichment of glycopeptides. Finally, the enriched sample was separated into eight high pH fractions and simultaneous discovery and reporter quantification were performed using LC-MS/MS. **b** For the assessment of site occupancy, both total cell lysates and secreted material were obtained from three technical replicates of wild type and *COSMC* knockout cells. Both sample types were reduced, alkylated, and digested with trypsin followed by *N*-glycan removal using PNGase F. Each sample was then separated into four high pH fractions and analyzed using LC-MS/MS, where relative occupancies were calculated based on MS1 peak intensity from glycosylated peptides and their non-glycosylated equivalents.

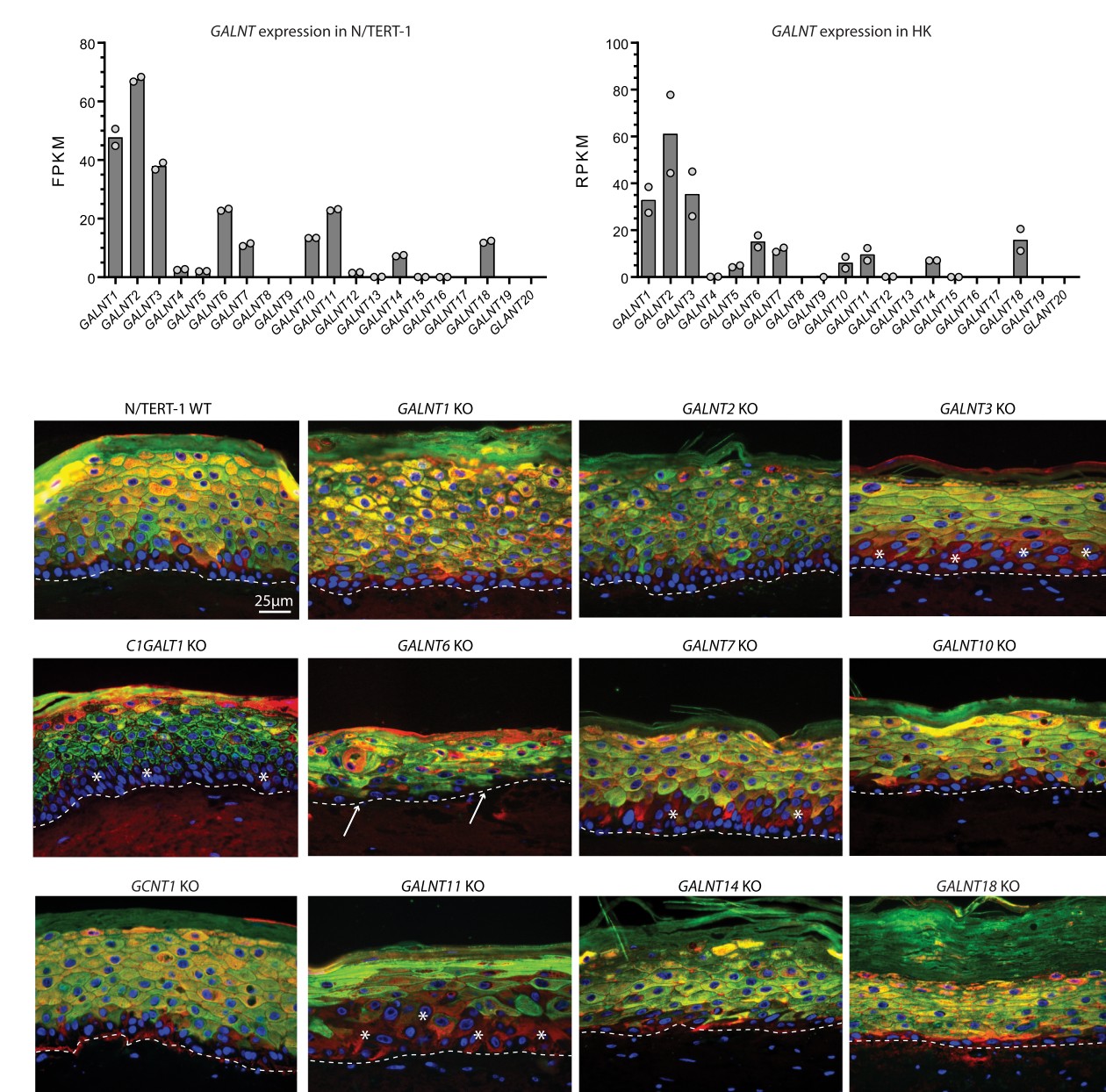

**Fig. 2 | GalNAc-T expression in N/TERT-1 cells and organotypic cell cultures.**
**a** RNA-Sequencing transcriptomics of N/TERT-1 cells[19] and primary human keratinocytes (HK)[20]. The difference in reported units of RPKM and FPKM is due to differences in sequencing method (single-end or paired end), but relative expression levels for GalNAc-Ts are similar between N/TERTs and primary human keratinocytes. Bars represent average values of two technical replicates. Source data are provided as a Source Data file. **b** Micrographs show sections of organotypic cultures made with N/TERT-1 WT and knockout (KO) cells grown on a collagen gel with embedded fibroblasts. All KO cells can form a stratified epithelium. *GALNT3*,

*GALNT7*, and *GALNT11* KO organotypic cultures show delayed differentiation (asterisks). *GALNT6* and *GALNT10* KO organotypic cultures shows a thinner epithelium, and *GALNT6* shows premature differentiation visualized by K10 staining cells at the basal layer (arrows). *GALNT18* KO generates an epithelium with a very thick stratum corneum. Sections are stained with the differentiation markers Keratin10 (green) and Involucrin (red) and nuclei are labeled with DAPI (blue). At least three independent experiments at different time points with three different clones of each knockout cell line were conducted, resulting in organotypic cultures with similar morphologies (Supplementary Fig. 1). Scalebar = 25 μm.

## Results

### Loss of individual GalNAc-Ts has a differential effect on tissue formation in a human 3D organotypic skin model

As previously reported, N/TERT-1 cells express a large proportion of the GalNAc-T family members[19], including isoforms T1, T2, T3, T6, T7, T10, T11, T14, and T18, which matches the transcriptomic profile of human primary keratinocytes[20] (Fig. 2a). Based on this, we targeted all nine GalNAc-Ts individually for genetic knockout. Three distinct clones from each knockout were expanded. The desired genetic change was confirmed using indel-detection by amplicon analysis (IDAA)[21] and DNA sequencing (Supplementary Data 1). We have

previously used N/TERT-1 cells to form human skin in a 3D organotypic model to assess the biological significance of individual glycosyltransferases during human tissue formation[19]. Here, we applied this strategy and evaluated the phenotypic consequences of the loss of each of the GalNAc-Ts expressed in human skin. As expected, the wild-type cells form a well-organized, stratified, and keratinized epithelium resembling normal human skin and consisting of 8-10 cell layers with a single basal cell layer negative for the differentiation marker K10[19]. In contrast, tissues formed with keratinocytes ablated for each of the expressed GalNAc-Ts caused, for the most part, changes to the overall differentiation pattern and tissue architecture (Fig. 2b). While lack of

GalNAc-T1 and GalNAc-T2 only caused minor changes to the cellular morphology (Fig. 2b and Supplementary Fig. 1), deficiency of GalNAc-T3 caused a delayed differentiation as visualized by two or more basal cell layers without K10 expression (Fig. 2b and Supplementary Fig. 1, asterisks). An opposite pattern was observed for GalNAc-T6 knockout tissue, exhibiting early activation of the differentiation program with premature basal expression of K10 (Fig. 2b). This is in accordance with our previous finding that GalNAc-T6 is essential to prevent differentiation in the colonic cancer cell line, LS174T[22]. Finally, we observed differentiation defects and morphological changes in tissues lacking GalNAc-T7, GalNAc-T10, GalNAc-T11, GalNAc-T14, and GalNAc-T18, with the most pronounced effect observed in GalNAc-T18 knockout tissue. Here, the basal epithelium was flattened and dysmorphic, with maturation of the upper layers into a thickened stratum corneum, potentially due to impaired ability to shed differentiated cells (Fig. 2b and Supplementary Fig. 1). Interestingly, the GalNAc-T knockout phenotypes were more pronounced than the rather mild phenotypes observed in tissues formed with *C1GALT1* knockout and *GCNT1* knockout cells, preventing the formation of all core-1 and most core-2 O-glycan structures, respectively (Fig. 2b and Supplementary Fig. 1). While *GCNT1* knockout tissue more or less resembled wild-type tissue models, the loss of C1GalT1 resulted in mild changes of the tissue architecture with delayed differentiation of the basal cell layers (Fig. 2b). These results suggest that select loss of site-specific O-glycosylation has a more pronounced effect on tissue formation than loss of all elongated O-glycans.

## O-GalNAc glycosylation sites are abundant throughout the N/TERT-1 proteome

Having established the biological importance of site-specific glycosylation, we next mapped the location of the individual O-glycosylation sites across the proteome of N/TERT-1 cells. Three technical replicates of the N/TERT-1 wild type cells, as well as three different clones of the individual GalNAc-T KOs, were subjected to an isobaric labeling-based quantitative glycoproteomics approach[23] and lectin-based glycopeptide enrichment (Fig. 1a). Combining the results of all the cells analyzed, a total of 2890 O-glycosylation sites (serine, threonine or tyrosine residue occupied by an O-GalNAc glycan) were identified on 835 distinct glycoproteins (Fig. 3a and Supplementary Data 2). From all the sites, 40.5% (1172 sites) could be assigned to one of five general protein regions within the N/TERT-1 expressed proteome (including: stem regions, linker regions, functional regions, annotated protein domains and regions in protein termini), suggesting that a large set of the O-glycosylation sites is localized on unstructured areas or areas without any prior functional annotation. Among the sites localized on specific domains, we found that 11.4% (330) were located in the stem-regions (within 50 amino acids of transmembrane domain) of type-I and -II transmembrane proteins. This is consistent with previous findings[6,7,24] and particularly interesting given the importance of O-glycans to protect from protein cleavage[10,25] and to assist in structural stability and protrusion of functional termini projecting out from the cellular membrane[3]. Another 5.4% (155) mapped to regions in between domains (<100 AAs) again suggesting to provide protection, and 6.6% (192) mapped to protein termini, defined as 10 amino acids from the N- or C-terminal after potential signal peptide cleavage. Functional regions, which are defined as sequences in a protein with specific functional annotation (e.g. proteolytic cleavage, heparin binding or organelle trafficking) comprise 5% (142) of the sites. Finally, 14.2% (397) of the O-glycosylation sites mapped to specific structural protein domains as annotated in the UniprotKB database[26]. The domain types with most O-glycans were fibronectin type-III (49), Ig-like C2-type (39), thioredoxin (36), LDL-receptor class A (33), cadherin (26), and Ig-like V-type (20) domains (Fig. 3b). For the vast majority of these, only one O-glycosylation site was identified per individual domain (Fig. 3c), while some domains, such as the thioredoxin domains, were consistently decorated with several glycans. The thioredoxin domains are also the most consistently glycosylated, as we found 68% of all thioredoxin domains to have one or more O-glycans (Fig. 3d). Other commonly glycosylated domains include Ig-like V-type domains (58%) and ricin B-type lectin domains (50%), while EGF-like domains (0.8%) and Laminin EGF-like domains (0.9%) are unlikely to be glycosylated. Finally, we found a trend showing that sites are preferentially located around the outer edges of the domains (Fig. 3e). This pattern is especially clear on specific domains such as Ig-like C2-type domains, LDL-receptor class A domains and Ig-like V-type domains, while other domains such as cadherin, thioredoxin and fibronectin type-III have a more uniform distribution of sites throughout the domain sequence.

## The N/TERT-1 proteome contains patches with high O-GalNAc glycosylation density

High-density glycan patches often have distinct biological functions, as exemplified by the canonical mucins. To enable the interrogation of O-glycan functions localized to mucin-like regions, we next created a comprehensive list of proteins containing high density O-glycan patches within the N/TERT-1 proteome. We defined high density glycan patches as being areas with 5 or more identified O-glycosylation sites within a stretch of 25 amino acids (Fig. 4a and Supplementary Data 3). Of the 92 identified O-glycan patches, the two patches with the highest glycan density both contained 11 O-glycosylation sites and were found in the stem-region of LDLR and on the extracellular domain of the collagen alpha-1(XVII) chain. Patches with seven or more potential O-glycosylation sites are illustrated in Fig. 4b. Fifteen of the theoretical patches were found in the top-30 most identified multi-site (≥4) glycoPSMs (represented by 26 different glycopeptides; Fig. 4c). A well described function of glycan patches is their role in protection from ectodomain-shedding of membrane protein stem-regions. Several glycan patches were indeed found in the stem-region of proteins such as CD44, LDLR, and ephrin-B1. While single-site O-glycans have the potential to serve as biological on/off switches through regulation by specific GalNAc-Ts, this is less likely to be the case for the sites in a mucin-like domain, which can be glycosylated by different GalNAc-Ts.

## A subset of the global O-GalNAc glycoproteome is regulated by specific GalNAc-T isoforms

Having established the global distribution of O-glycosylation sites, we next analyzed the contribution of the individual GalNAc-Ts to the N/TERT-1 O-glycoproteome (Fig. 1a). To interpret site-specific differential changes in the *GALNT* knockout glycoproteomes, we selected the 1579 sites (54.6% of the total) represented by glycopeptides carrying a single glycosylation site and present on proteins that did not change in abundance between the knockouts and the wild type (Fig. 5a and Supplementary Data 4). Of these, we found 325 glycosylated sites (11.2%), of which 92% was unambiguously assigned to a specific amino acid in the peptide, with a significantly lower abundance in at least one of the *GALNT* knockouts, as compared to the wild-type cells (fold change < 0.5 and False Discovery Rate (FDR) adj. *p*-value < 0.05). Furthermore, 281 sites (9.7%) were found to be specifically regulated by only one of the nine GalNAc-T isoforms addressed in this study. The limited proportion of isoform-regulated sites is in good agreement with the overlapping specificities of GalNAc-Ts[3,4,27], and provides a focused list of prospective protein candidates, functions of which are likely to be tuned by individual GalNAc-Ts.

For GalNAc-T11 a very strong effect, limited to the glycan sites of LDLR and LRP1, was observed, which corresponded well with the known substrate selectivity of this transferase[11,28]. The glycosylation shift in the glycoproteomes of the other GalNAc-T knockouts varied both in terms of the quantity of affected sites and in the magnitude of the changes. In the knockouts of GalNAc-T1, -T2, -T3, and -T6 we found the highest numbers of downregulated glycosylation sites, showing a fold change between 0.5 and 0.03 (Fig. 5c). The knockout of GalNAc-T2

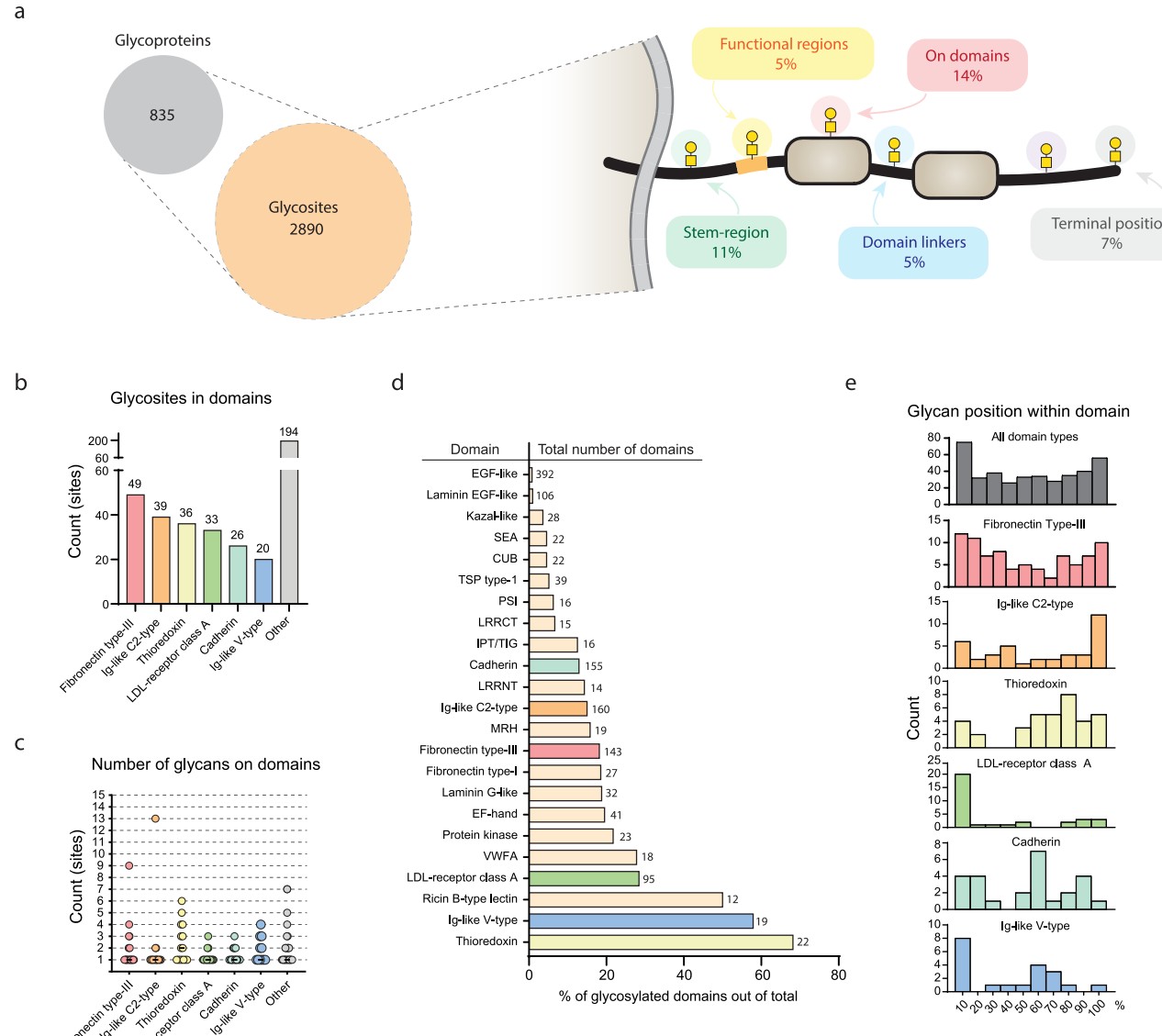

**Fig. 3 | O-GalNAc glycans on protein domains. a** Glycoproteomic analysis of N/TERT-1 cells revealed a total of 2890 O-GalNAc sites on 835 glycoproteins. The sites found were mapped to distinct parts of proteins. Sites that do not map to distinct protein features (purple) account for 59% of the total sites **b** Number of O-GalNAc sites found in different types of domains (top 6). **c** Number of O-GalNAc sites on individual domains. Each dot represents a specific domain with one or more sites. Black bars show median number of sites per domain. **d** Glycosylation rate of each domain calculated as domains with at least one site, divided by total number of domains. Numbers on the right of each bar shows total number of specific domain type within the 835 identified glycoproteins. **e** Relative glycan positions within each domain. Each bin covers 10% of domain sequence from N- to C-terminus. Top 6 domain types from **b** are color highlighted throughout. Source data are provided as a Source Data file.

resulted in the most dramatic effect on the O-glycoproteome, which is in accordance with findings of GalNAc-T2 knockouts in HaCaT keratinocytes[29]. Upon Gene Ontology (GO)-term analysis (Supplementary Fig. 2) of GalNAc-T2 target proteins, a strong association with cell adhesion was recapitulated. GalNAc-T2 regulates sites on components of hemidesmosomes (COL17A1, laminins), adherens junctions (nectins), desmosomes (DSG3), tight junctions (F11R), as well as focal (ITGA5, ITGB6) and fibrillar adhesions (ITGA5). Of those, sites on integrins and laminins were, to our knowledge, for the first time identified as GalNAc-T2-specific substrates. The Ser[901] glycosite on ITGA5 may have regulatory potential, as it is located close to the PCSK5 cleavage site required for precursor processing into the ITGA5 heavy and light chains and thus activation of the protein. Among other associations, GalNAc-T2 selectively glycosylated subsets of proteins involved in cellular redox homeostasis and response to hypoxia, angiogenesis, and cell migration. Compared to previously published

*GALNT2* knockout cell lines[29–31], we similarly found ubiquitous proteins ROBO1 (Thr[660]) and MIA3/TANGO1 (Thr[768]) to be specific substrates of GalNAc-T2. Other examples of previously reported GalNAc-T2 substrates found in this study are proteases PRCP (Thr[39]) and CTSD (Thr[63]), with both sites situated in close proximity to propeptide processing sites, as well as endocytic receptors LRP1 (Thr[3936]) and LSR (Ser[546]). We also found a GalNAc-T2-specific site on endothelial lipase (LIPG; Thr[41]), a main regulator of blood HDL-C levels, which is of particular interest given the many reports on GalNAc-T2s involvement in HDL-C metabolism[16,32,33].

GO-term analysis of GalNAc-T1 target proteins revealed an association with the endomembrane components involved in vesicular transport previously seen in keratinocytes[29]. Furthermore, several secretory pathway targets were consistently found for GalNAc-T1 and included ERLIN2 (Thr[336]), SLC9A1 (Thr[62]), TGOLN2 (Thr[202]), TXNDC5 (Thr[174]), PRKCSH (Thr[263]), and HYOU1 (Ser[634]; Thr[881]) suggesting an

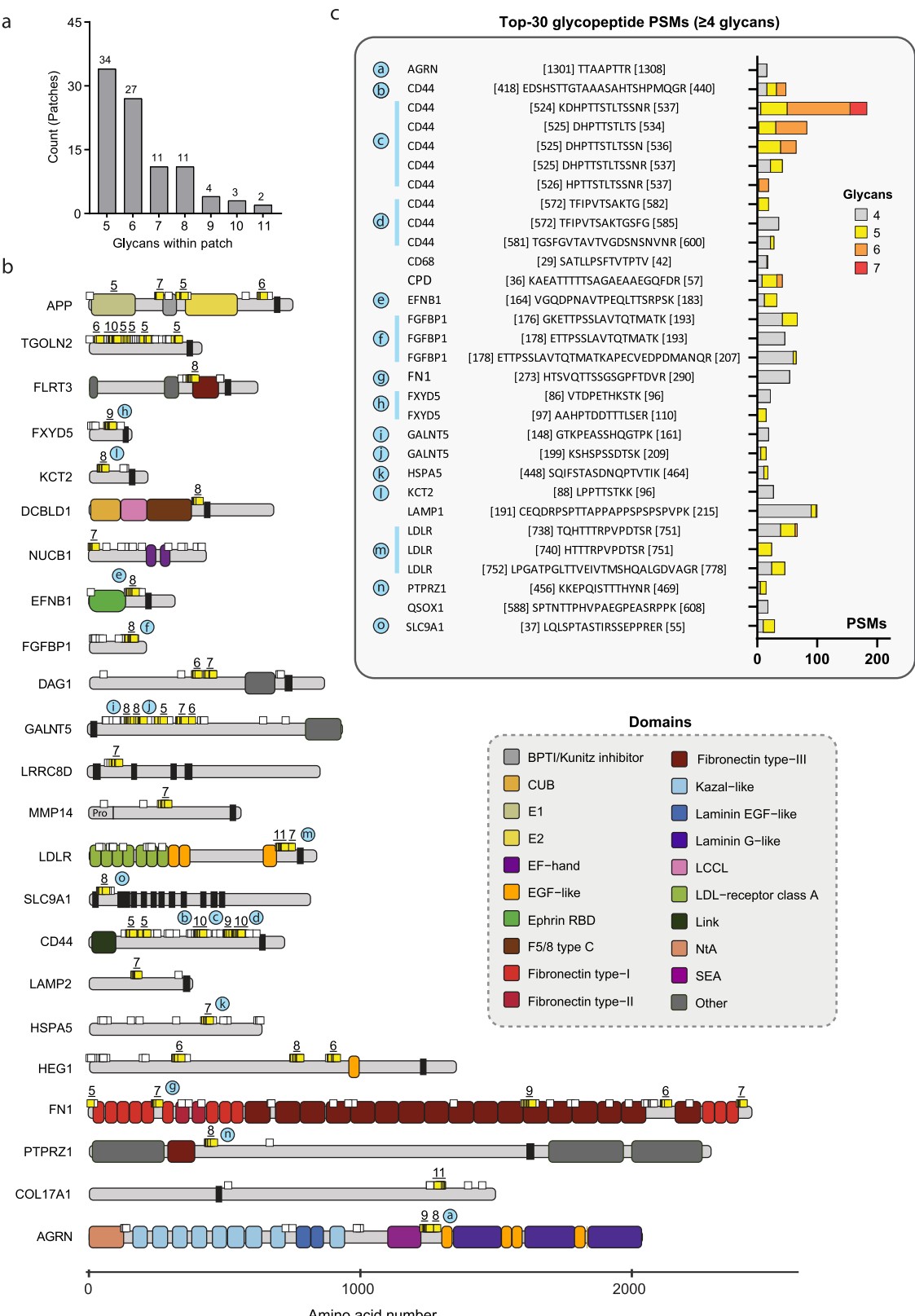

**Fig. 4 | Dense O-GalNAc patches in the N/TERT-1 glycoproteome. a** Number of patches (≥5 sites within a 25 amino acid interval) with different number of O-GalNAc sites. **b** Glycoproteins with theoretically mapped glycan patches of 7 sites or more (yellow). Other sites not in patches are shown in white. **c** Top 30 experimentally identified glycoPSMs with ≥4 glycans. Letter-coded blue circles show mapping of top 30 glycoPSMs (**c**) to the theoretical patches (**b**). Source data are provided as a Source Data file.

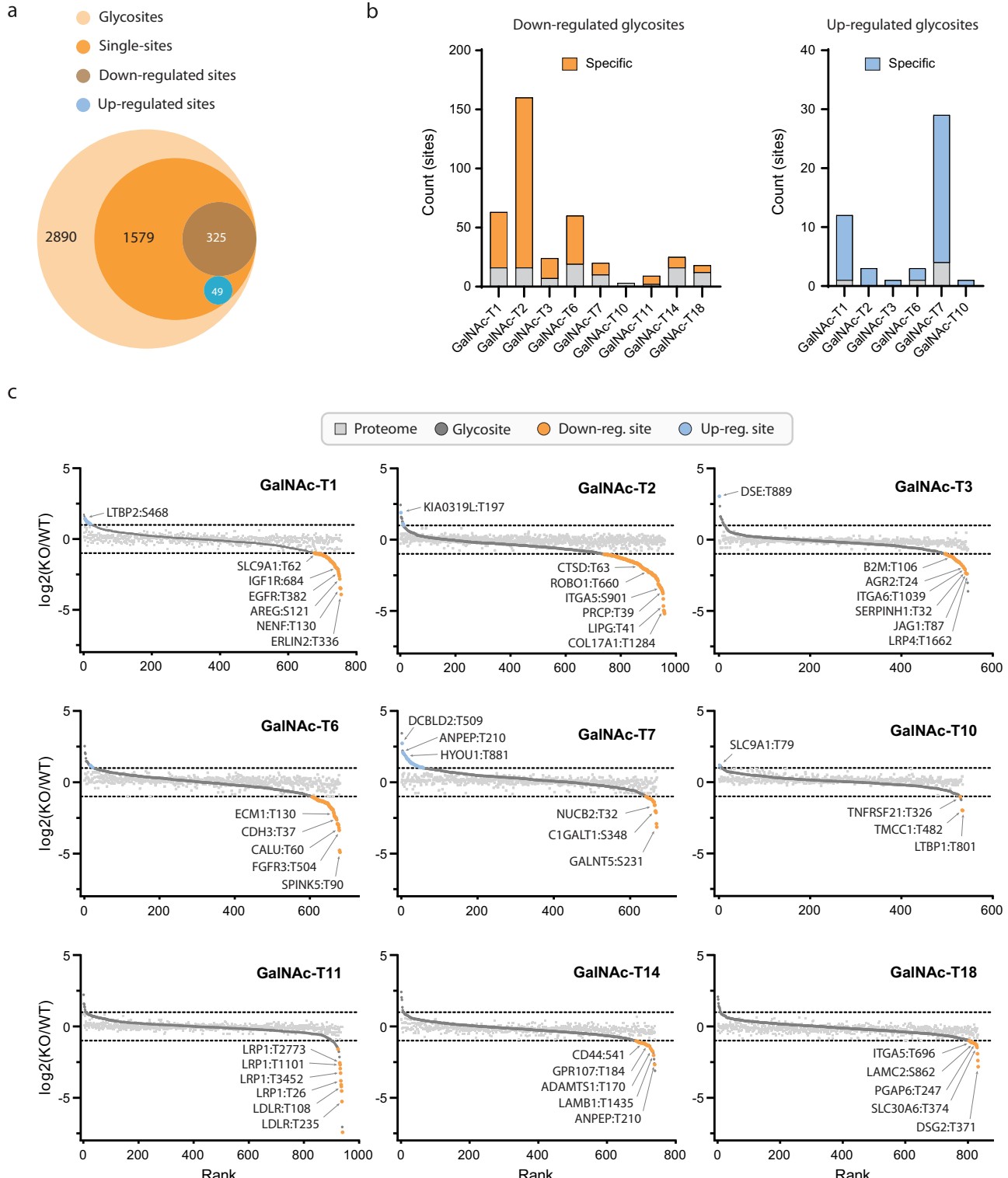

**Fig. 5 | GalNAc-T isoform-specificity. a** Out of the 2890 glycosites 1579 sites were selected being single-glycan glycopeptides. Overall, 325 sites (brown circle) were significantly less abundant in one or more of the nine *GALNT* knockouts and 49 sites (blue circle) were significantly more abundant. **b** Number of significantly regulated sites in each *GALNT* knockout. Isoform-specific sites highlighted in orange (down) or blue (up). Source data are provided as a Source Data file. **c** Rank curve plot showing change in glycan site (dark gray circles) abundance for each of the 9 *GALNT* knockouts. Orange circles represent significantly down-regulated sites (*p*-value < 0.05 and log2 fold change < −1) and blue circles represent

significantly up-regulated sites (*p*-value < 0.05 and log2 fold change > 1). Sites from proteins with overall abundance change in the proteome (light gray squares) were removed. For each of the nine targeted *GALNT* genes, knockouts (*n* = 3 biologically independent samples) were compared in a multiplexed experiment with wild-type material (*n* = 3 technically independent samples). A two-sided Student's *t* test was used as a measure of statistical confidence and the *p*-values were adjusted for multiple testing, using an FDR of 5%. Source data are in Supplementary Data 2 and 4.

ubiquitous role of this GalNAc-transferase in the secretory pathway. However, GalNAc-T1 may also exert tissue-specific regulatory functions on autocrine and paracrine signaling, as several receptor tyrosine kinases harbored GalNAc-T1-specific sites, including IGF1R (Thr[684]) and EGFR (Thr[382]). Furthermore, we found regulated sites on growth factors AREG (Thr[169]), as well as one of the Wnt receptors FZD6 (Thr[150]), and also replicated GalNAc-T1-specific sites on AREG (Ser[121]), NENF (Thr[130]), and SPINT2 (Thr[101]) previously found in HaCaT keratinocytes.

The GalNAc-T3-specific dataset generated a limited number of unique regulated sites. Of those, JAG1 (Thr[87]), AGR2 (Thr[24]), LRPAP1 (Thr[134]), and B2M (Thr[106]) were a replication of the HaCaT differential dataset[29]. Interesting sites include those found on integrins (ITGA6 (Thr[1039]) and ITGB1 (Thr[199])) and laminins, which are distinct from those found in the *GALNT2* knockout dataset. We did not find any overlap between GalNAc-T6-specific targets found in this study and previously published *GALNT6* knockout data from another cellular system[22]. The GO-term analysis based on GalNAc-T6 target proteins revealed involvement in wound healing and association with ECM components such as proteoglycans and secreted factors. Of the identified GalNAc-T6 targets, a few worth mentioning include adhesion molecules CDH3 (Thr[37]), ECM1 (Thr[130]) and ITGAV (Thr[888]). Furthermore, GalNAc-T6 selectively glycosylated several sites on proteins involved in wound repair such as EFNB1 (Thr[211]) and SDC1 (Thr[115]; Thr[122]; Thr[220]).

As O-glycosylation is an abundant modification of secreted proteins[6], we evaluated the overlap between our results in the total cell lysates and the differential glycoproteomics of secreted material from GALNT1 and GALNT2 knockouts (Supplementary Fig. 3 and Supplementary Data 2). We found that the majority of glycoproteins (86%) and glycosites (77%) identified in the secretome samples were covered by the total cell lysate approach, which suggests that the enrichment strategy effectively picks up glycopeptides from secreted glycoproteins in total cell lysate samples. The secretome added 32 new glycoprotein identifications (3.7% of total) and 133 new glycosylation site identifications (4.4% of total) as compared to the total cell lysate, of which 14 were regulated by either GalNAc-T1 or -T2. The knockout of GalNAc-T1 had a relatively small impact on the glycosecretome, while the impact of the knockout of GalNAc-T2 was substantial. Additional T2-specific sites found in the secretome included PXDN (Thr[1348]) in the region important for homotrimerization of the protein, and on the propeptide of MMP1 (Thr[23]). Important T2-regulated sites already found in the lysate samples, including PRCP (Thr[39]), LIPG (Thr[41]), CTSD (Thr[67]), and JAG1 (Thr[904]), were confirmed in the glycosecretome.

For the GalNAc-T7, -T10, -T14, and -T18 knockouts a limited number of significantly altered glycosylation sites were found, and the magnitude of most of these changes was comparatively small. This can be partly explained by some of these enzymes (GalNAc-T7 and -T10) being late-acting GalNAc-Ts, responsible for follow-up glycosylation of previously glycosylated regions. This was indeed reflected in the analysis of the multi-glycosylated peptides in our data (Supplementary Data 2), where additional T7-regulated regions were found for e.g. the proteins FGFBP1, AGRN, ERP44, and COL17A1. We also observed a limited number of up-regulated glycosylation sites in some of the GalNAc-T knockouts (Fig. 5b). The majority of these sites were found in GalNAc-T1 and -T7, and are likely caused by the loss of follow-up glycosylation of prior glycosylated sites represented by peptides that can carry multiple sites. As expected, the extent of this effect was more apparent when including multi-site glycopeptides in the analysis (Supplementary Fig. 4).

## O-glycosylation site occupancy is variable throughout the proteome

To better determine the importance of each O-glycosylation site, we next aimed to address the occupancy of the sites under native conditions. Inherently to the applied glycopeptide enrichment strategy this information is lost in the current differential glycoproteomics

approach. To assess the O-glycosylation site occupancy we therefore performed a MS1-based relative quantification of glycopeptides and their non-glycosylated equivalents in sample material without the specific enrichment of glycosylated forms of proteins or peptides (Fig. 1b). To this end we investigated N/TERT-1 cells with a knockout of the *C1GALT1C1* gene (*COSMC*). COSMC is essential for elongation of O-glycans with galactose beyond the initial GalNAc-residue and its deletion simplifies the repertoire of O-glycans, as well as reduces the possible bias between the quantification of the peptides and their glycosylated variants[34]. Next to the total cell lysate, we used the secreted material from the *COSMC* knockout cells, which does not contain the non-glycosylated pool of nuclear and cytosolic proteins. For these two sources we were able to perform relative MS1-based quantification of 95 unique glycosylated peptide precursors, covering 70 unique glycosylated regions in the proteome, and their non-glycosylated counterparts (Fig. 6a and Supplementary Data 5). As simplification of the O-glycosylation pathway might influence the O-glycan occupancy at individual sites via enhanced follow-up glycosylation[3], we searched for counterparts of the quantified glycopeptides with more elaborated glycans in wild-type samples. Due to the tremendous complexity of wild-type O-glycosylation[35], for the wild-type material we were only able to make confident MS1 quantifications on single-site glycopeptide precursors. This resulted in quantification of 31 unique glycosylated precursors with 10 different glycan compositions, covering 23 unique protein regions. For the glycosylation sites quantified in both wild type and *COSMC* knockout samples, the ratio between the glycosylated and non-glycosylated variants of the same peptide sequence correlated well between the two sources ($R^2$: 0.80; Fig. 6b), suggesting that the larger dataset obtained from *COSMC* knockout material can be used to approximate occupancy across the N/TERT-1 proteome.

The complete set of identified pairs of glycosylated and non-glycosylated variants from different protein targets in *COSMC* knockout samples showed a large variation in O-glycan occupancies between different glycosylation sites on different proteins, while replicates of the same site were consistent. Overall, ~60% of the sites were highly occupied (ratio between glycosylated and non-glycosylated peptide >0.75), while 11% showed a ratio between 0.25 and 0.75, and close to 30% of the sites displayed low to very low occupancy (between 0.1 and 0.001). The different levels of O-glycan occupancy were also observed for different glycosites within the same protein, as exemplified by fibronectin (FN1) and nucleobindin-1 (NUCB1; Fig. 6c and Table 1). On FN1, glycopeptides from the N- and C-termini and the linker region between FN-Type-I domain 5 and 6 display a medium occupancy (0.16–0.42), while most of the sites on FN-Type-III domains have very low occupancy (<0.05). Interestingly, the three consecutive glycopeptides in the variable region showed a moderate to high occupancy (0.45–0.99) suggesting their functional importance. Also glycoproteins that carry sites specifically regulated by individual GalNAc-Ts showed different degrees of occupancy (Fig. 6c and Table 1). The three GalNAc-T11-specific sites in LDLR were exclusively covered by glycopeptides, and the non-glycosylated forms not identified, assumingly indicating high occupancy. Likewise, we found close to complete occupancy of the extracellular matrix protein 1 (ECM1) glycopeptide with the GalNAc-T6 site Thr[130]. Furthermore, the GalNAc-T2-specific site Thr[754] on agrin (AGRN) showed high occupancy (0.96 ± 0) and we found a medium occupancy (0.31 ± 0.02) on the cadherin-3 (CDH3) peptide that covers the GalNAc-T6-specific Thr[37] site. Low occupancy was found for the Jagged-1 protein (JAG1; 0.03 ± 0) and urokinase-type plasminogen activator (PLAU; 0.10 ± 0.01) peptides that contain the GalNAc-T3-specific Thr[87] site and GalNAc-T2-specific Ser[158] site, respectively. The glycan patches described earlier for the stem region of LDLR and APP, the regions adjacent to the SEA domain in AGRN, and several peptides from CD44 (Fig. 4b, c) were found with high occupancy (Fig. 6c and Supplementary Data 5). In conclusion, targeted

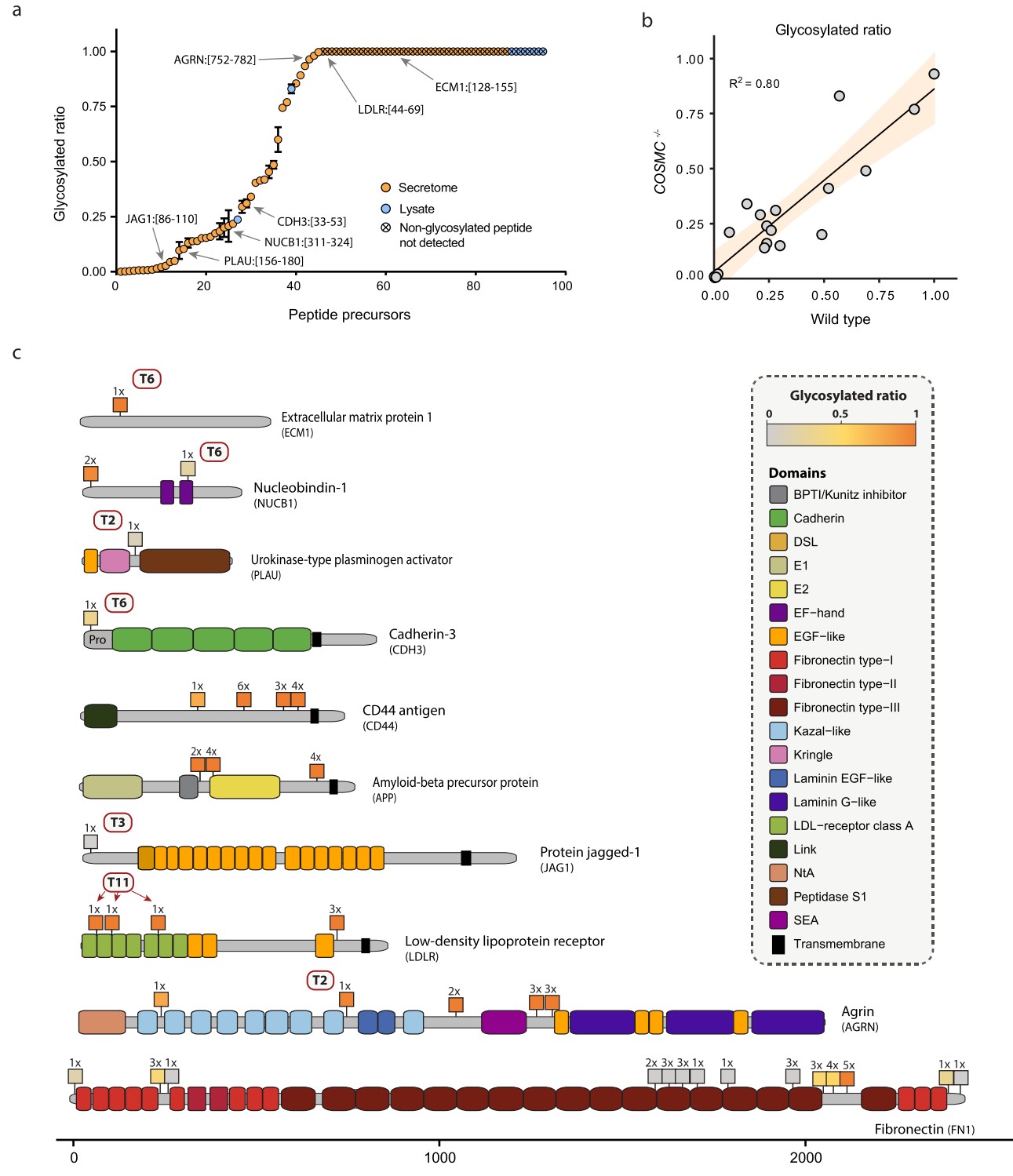

**Fig. 6 | O-glycosylation site occupancy on N/TERT-1 proteins. a** MS1-based relative quantification of O-glycosylation site occupancy on N/TERT-1 *COSMC*[−/−] peptides from both lysate (blue) and secreted protein (orange). Crossed data points show glycopeptides for which the non-glycosylated equivalent was not detected and the occupancy was assumed to be high. Highlighted peptides correspond to isoform-specific sites as shown in subfigure **c**. Source data are in Supplementary Data 5. **b** Comparison of occupancy data derived from single-site glycopeptides from both wild type and *COSMC* knockout samples. Linear correlation line plotted ($R^2 = 0.80$) with 95% confidence region. Source data are provided as a Source Data file. **c** Graphical illustration of site occupancy on proteins of interest from *COSMC* knockout material. Color intensity of squares reflect the ratio between the glycosylated and the non-glycosylated peptides as indication for site occupancy. Numbers above squares indicate the max. number of O-GalNAc glycans on the quantified peptide. Red boxes show isoform-specific sites (e.g. T2 for GalNAc-T2). All data in the figure represent averages of technical replicates ($n = 3$). Measures of technical variation can be found in the form of standard deviations represented by error bars (**a**), or in Supplementary Data 5.

**Table 1 | Relative O-glycan site occupancies in proteins from N/TERT-1 COSMC knockout cells**

| Protein | Accession | Peptide | Tn max | Glyco ratio[a] ±SD | xTn (%) 1 | 2 | 3 | 4 | 5 | 6 |
|---|---|---|---|---|---|---|---|---|---|---|
| CDH3 | P22223 | [33]EAEVTLEAGGAEQEPGQALGK[53] | 1 | 0.31±0.01 | 100 | - | - | - | - | - |
| NUCB1 | Q02818 | [32]GAPNKEETPATESPDTGLYYHR[53] | 2 | 0.93±0 | 84 | 16 | - | - | - | - |
| NUCB1 | Q02818 | [311]LVTLEEFLASTQRK[324] | 1 | 0.2±0.05 | 100 | - | - | - | - | - |
| JAG1 | P78504 | [86]VTAGGPCSFGSGSTPVIGGNTFNLK[110] | 1 | 0.02±0 | 100 | - | - | - | - | - |
| PLAU | P00749 | [156]KPSSPPEELK[180] | 1 | 0.1±0.01 | 100 | - | - | - | - | - |
| LDLR | P01130 | [44]WVCDGSAECQDGSDESQETCLSVTCK[69] | 1 | High[b] | 100 | - | - | - | - | - |
| LDLR | P01130 | [108]TCSQDEFR[115] | 1 | High[b] | 100 | - | - | - | - | - |
| LDLR | P01130 | [224]DKSDEENCAVATCRPDEFQCSDGNCIHGSR[253] | 1 | High[b] | 100 | - | - | - | - | - |
| LDLR | P01130 | [710]SCLTEAEAVATQETSTVR[728] | 4 | High[b] | - | - | 41 | 58 | - | - |
| CD44 | P16070 | [314]AFDHTK[319] | 1 | 0.74±0.01 | 100 | - | - | - | - | - |
| CD44 | P16070 | [441]TTPSPEDSSWTDFFNPISHPMGR[463] | 6 | High[b] | - | - | - | 9 | 78 | 12 |
| CD44 | P16070 | [572]TFIPVTSAK[580] | 3 | High[b] | - | - | 100 | - | - | - |
| CD44 | P16070 | [581]TGSFGVTAVTVGDSNSNVNR[600] | 4 | High[b] | - | - | 60 | 40 | - | - |
| AGRN | O00468 | [240]GPCGSRDPCSNVTCSFGSTCAR[261] | 1 | 0.77±0.01 | 100 | - | - | - | - | - |
| AGRN | O00468 | [752]GPTFAPLPPVAPLHCAQTPYGCCQDNITAAR[782] | 1 | 0.96±0 | 100 | - | - | - | - | - |
| AGRN | O00468 | [1026]TTASVPR[1032] | 2 | High[b] | - | 100 | - | - | - | - |
| AGRN | O00468 | [1279]LPSSAVTPR[1287] | 3 | High[b] | - | - | 100 | - | - | - |
| AGRN | O00468 | [1288]APHPSHTSQPVAK[1300] | 3 | High[b] | - | - | 100 | - | - | - |
| APP | P05067 | [352]TTQEPLAR[359] | 2 | High[b] | - | 100 | - | - | - | - |
| APP | P05067 | [360]DPVKLPTTAASTPDAVDKYLETPGDENEHAHFQK [393] | 4 | High[b] | - | - | - | 100 | - | - |
| APP | P05067 | [649]GLTTRPGSGLTNIKTEEISEVK[670] | 5 | High[b] | - | 1 | 33 | 62 | 2 | - |
| ECM1 | Q16610 | [128]EGTPAPFGDQSHPEPESWNAAGHCQQDR[155] | 1 | High[b] | 100 | - | - | - | - | - |
| FN1 | P02751 | [32]QAQQMVQPQSPVAVSQSKPGCYDNGK[57] | 1 | 0.15±0.01 | 100 | - | - | - | - | - |
| FN1 | P02751 | [273]HTSVQTTSSGSGPFTDVR[290] | 3 | 0.42±0.01 | 29 | 32 | 39 | - | - | - |
| FN1 | P02751 | [291]AAVYQPQPHPQPPPYGHCVTDSGVVYSVGMQWLK[324] | 1 | 0.01±0 | 100 | - | - | - | - | - |
| FN1 | P02751 | [1631]TEIDKPSQMQVTDVQDNSISVK[1652] | 2 | 0.01±0 | 12 | 88 | - | - | - | - |
| FN1 | P02751 | [1653]WLPSSSPVTGYR[1664] | 3 | 0.02±0 | - | 53 | 47 | - | - | - |
| FN1 | P02751 | [1665]VTTTPKNGPGPTK[1677] | 4 | 0.05±0.01 | - | 2 | 9 | 89 | - | - |
| FN1 | P02751 | [1728]GLAFTDVDVDSIK[1740] | 1 | 0.01±0.01 | 100 | - | - | - | - | - |
| FN1 | P02751 | [1821]FTQVTPTSLSAQWTPPNVQLTGYR[1844] | 1 | 0.01±0.01 | 100 | - | - | - | - | - |
| FN1 | P02751 | [2002]FLATTPNSLLVSWQPPR[2018] | 3 | 0.002±0 | - | - | 100 | - | - | - |
| FN1 | P02751 | [2080]KTDELPQLVTLPHPNLHGPEILDVPSTVQK[2110] | 3 | 0.41±0 | 34 | 22 | 44 | - | - | - |
| FN1 | P02751 | [2011]TPFVTHPGYDTGNGIQLPGTSGQQPSVGQQMIFEEHGFR[2149] | 4 | 0.60±0.05 | 51 | 36 | 8 | 5 | - | - |
| FN1 | P02751 | [2059]RTTPPTTATPIR[2070] | 5 | 0.98±0 | 14 | 16 | 10 | 47 | 11 | - |
| FN1 | P02751 | [2060]TTPPTTATPIR[2070] | 5 | 0.99±0 | 51 | 21 | 9 | 13 | 3 | - |
| FN1 | P02751 | [2426]RPGGEPSPEGTTGQSYNQYSQR[2447] | 2 | 0.22±0.01 | 91 | 9 | - | - | - | - |
| FN1 | P02751 | [2452]TNTNVNCPIECFMPLDVQADREDSRE[2477] | 1 | 0.001±0 | 100 | - | - | - | - | - |

[a]Average glycosylated ratio (glyco ratio), standard deviation (SD), and the relative abundance of different numbers of Tn residues on the peptide (xTn; %) were calculated over three technical replicates. [b]Non-glycosylated peptide not detected.

quantitative analysis of select glycosylated and non-glycosylated peptide pairs revealed a wide range of O-glycosite occupancies across different proteins and domains, potentially narrowing down the data space for individual protein biology-focused follow up studies.

## Discussion

In this study, we mapped the glycosylation capacity of the nine expressed GalNAc-T isoforms in the N/TERT-1 keratinocyte cell line, in combination with the occupancy of a set of O-glycosylation sites, to identify O-GalNAc glycosylation sites with high functional potential. We have previously introduced the combined use of genetic engineering and tissue models to examine how complex N-glycans as well as extended glycosphingolipids and O-glycans are involved in differentiation during human tissue formation[19,22,29]. In the present study we applied this strategy to map site-specific O-glycosylation governed by GalNAc-Ts, using the established human skin cell line N/TERT-1, capable of forming human 3D organotypic skin tissue[19]. We created knock out cells for each of the nine GalNAc-Ts expressed in the N/TERT-1 cells and demonstrated a more profound and biologically significant effect of O-glycan initiation on epithelial formation compared to the first elongation step or branching. Here, it needs to be noted that the model only reflects the endogenous effects of glycans in differentiation and tissue-formation of the keratinocytes, and not the interaction with other cell types such as immune cells. The loss of complete glycans will most likely affect localization, surface expression and secretion of specific proteins, while loss of all elongated glycans and glycan decoration might play more significant roles in tissue stability under stressed conditions and in the interaction between different cell types[19]. We next used mass spectrometry to map the O-glycosylation sites in N/TERT-1, and found that O-glycans were largely located on unstructured regions, but also localized to protein regions and domains with known functional characteristics as well as on mucin-like domains. Subsequently, we examined the differential O-glycoproteomes for each of the nine expressed GalNAc-Ts, and show that the transferases exert discrete control of select substrates, consistent with their differential impact on the tissue phenotypes. Finally, to address one of the missing and essential questions in the field we analyzed the O-glycan occupancy of a subset of the O-glycosylated sites. We discovered large differences in O-glycan occupancy, ranging from very low to almost full occupancy; a finding that questions the relevance of the high number of O-glycosylation sites previously reported, and provides help for the selection of relevant sites which can be pursued in future functional studies.

Mapping of the global O-glycoproteome received considerable attention in the past decade, exemplified by the various studies reporting thousands of O-glycosylation sites in various cell systems[6–9,29–31]. Some approaches were based on genetically simplified glycosylation backgrounds, expressing only monomeric GalNAc or dimeric NeuAc-GalNAc glycans, which allows efficient, glycoform independent lectin enrichment of glycopeptides[6]. Others, like the current, rely on a native glycosylation machinery, and sialidase treatment, followed by the lectin-based enrichment of T structures[29]. Although the latter approach has the risk of missing sites that are exclusively occupied by O-glycan cores other than T antigen, it provides a more realistic view of the presence of sites in the native situation featuring glycan elongation. Complementary approaches to study the O-glycoproteome use O-glycosylation site-specific proteases to generate O-glycopeptides[8,36], allowing easy site identification in dense O-glycan clusters, or employ metabolic labeling combined with click-chemistry to enrich for O-glycopeptides[9,37]. Metabolic labeling approaches allow the targeting of a wider variety of glycoforms, but until now did not reach the depth of site identification as reported with lectin-based enrichments. Our data shows an overlap in identified glycoproteins between the keratinocytes in the current study and other cell types previously investigated

(including, amongst others, kidney, colon, liver, and blood derived material), but also a unique set of glycoproteins exclusively found in the current data (Supplementary Data 6).

To narrow down on the potential mechanistically relevant O-glycosylation sites, we performed an extensive characterization of the 397 identified sites within functional protein domains. We noted that in domains such as Ig-like C2-type, LDL-receptor class A, and Ig-like V-type, the majority of identified sites is located towards the edges of the annotated domain sequences. Whether the O-glycans found on the domain edges are directly tied to domain function, serve to stabilize the domain fold, or protect less structured areas towards linker regions has yet to be determined. The fact that the edge glycosylation is seen on different domain types with different functions could hint to a universal structural feature.

Another feature observed in the N/TERT-1 glycoproteome were patches with high density of O-glycans. Such patches are known as mucin domains, and form rigid and extended bottle-brush like structures[3,25] which serve to create biophysical protection from degradation and cleavage, and drive binding interactions with influence on receptor functions[3]. While the term "mucin domains" was originally reserved for the 21 canonical mucins[38,39], it is becoming increasingly clear that mucin domains, defined as high density O-glycan patches, exist across our entire proteome[40]. Out of the 68 proteins for which we identify glycan patches, 15 were described previously as having mucin-like domains[26,40], leaving 53 potentially newly described mucin domain identifications (Supplementary Data 3). The highly occupied mucin patches we identified on CD44, FN1, and APP were predicted to be mucin domains previously[40], while other identifications included FGFP1, EFNB1, and HSPA5. Glycan patches are expected to have distinct biological functions as the exact number of glycans in the patch can vary based on the activity of GalNAc-Ts, and thus serve as a functional gradient for e.g. interactions with glycan binding proteins such as Siglecs[41]. Glycan patches could also be directly involved in modulating protein-protein interactions, which is possibly the case for FGFBP1, where a glycan patch (S169-K193) is found directly adjacent to the FGF-binding site (K193-C234)[42]. We furthermore confirmed the presence of highly occupied glycan patches in the S/T-rich region flanking the C-terminal of the agrin SEA domain, which has long been associated with O-glycosylated areas[43], although the exact connection is still poorly understood. To further investigate the N/TERT-1 mucinome and its differential regulation by GalNAc-Ts, alternative protease cleavage regimes based on O-glycoproteases could be applied, which will likely increase the annotation of sites within mucin-like domains even further[41].

Next, we specifically assessed the glycosylation sites in the keratinocyte proteome that were differentially regulated by a specific GalNAc-T. As compared to previous studies[22,29–31,44], we here present a systematic interrogation of the complete panel of relevant GalNAc-T enzymes, using a tissue-forming keratinocyte model that is close in resembling true human biology. Additionally, our data, based on three biological replicates per condition, appeared highly robust, resulting in a high number of statistically significant altered sites in the different knockouts.

About one fourth of the quantified sites in this study were found to be affected in one or more of the *GALNT* knockouts, with *GALNT1, -2* and *-6* having the largest impact. For example, GalNAc-T1 glycosylated components are involved in EGFR and IGFR signaling, important for epidermal development. GalNAc-T2, which is primarily expressed in basal cells, selectively glycosylated cell-cell and cell-matrix adhesion molecules such as the integrins ITGA5 and ITGB6. In contrast, the integrins ITGA6 and ITGB1, involved in hemidesmosome formation and directional migration, respectively, were glycosylated by GalNAc-T3. Given the link between GalNAc-T6 and early epithelial transformation[22], it is interesting that *GALNT6* knockout tissues exhibited severe differentiation defects. As a potential explanation,

GalNAc-T6 specifically glycosylated a subset of adhesion and signaling molecules (including two ephrin ligands), different from those glycosylated by GalNAc-T1 and GalNAc-T2. Another essential protein harboring specific GalNAc-T6 glycosylation sites is the integrin ITGAV, which is often overexpressed in cancers and important for keratinocyte proliferation[45]. Interestingly, we also found a glycosylation site primarily regulated by GalNAc-T6 in ECM1, a protein contributing to the maintenance of skin integrity and homeostasis[46]. Of note, the glycosylation site in ECM1 (Thr[130]) was found to be fully occupied in wild-type material, and a short nucleotide polymorphism (SNP) affecting this amino acid has been associated with ulcerative colitis[47]. Ablation of *GALNT3, GALNT7, GALNT11*, and *GALNT18* resulted in differentiation phenotypes with various severity. Based on the largely non-overlapping and rather discrete substrate specificities between the four enzymes, the phenotypic consequences most likely result from an effect on distinct pathways. *GALNT11* presents an example of an enzyme which selectively glycosylates only few substrates, in this case from the LDLR-like superfamily[11,28,48], which is expected to allow for relatively straightforward functional dissection in future studies. The phenotype observed for the lesser-studied GalNAc-T18 is quite prominent, and also seems to be derived from the glycosylation of a low number of targets. GalNAc-T18, however, has so far no proven catalytic activity in vitro[49]. Studies have shown that GalNAc-T18 enhances the in vitro activity of GalNAc-T2 and -T10, and suggested that GalNAc-T18 has a chaperone-like effect rather than direct catalytic activity[50]. Indeed we found 10 of the GalNAc-T2 and 2 out of the 3 GalNAc-T10 targets to overlap with the GalNAc-T18 targets. This might explain the dramatic phenotype in *GALNT18* knock out tissues.

A few other differentially glycosylated sites warrant further attention. Thr[63] found on the lysosomal aspartyl protease cathepsin D (CTSD), composed of a protein dimer produced from a single protein precursor[29,51], was previously reported to be regulated by GalNAc-T2[29,51]. Here, we additionally find GalNAc-T6 to contribute to the glycosylation of this site. Interestingly, this particular site on CTSD is positioned on the propeptide directly adjacent to the pro-protein cleavage site ([57]AVPA**T**V↓EGPIPE[69]) necessary for generating a mature and active enzyme[52], indicating that the regulation of this site by GalNAc-T2 and/or -T6 might play a role in the maturation of the enzyme. Finally, although not expected to have a direct impact on the observed tissue phenotypes, the GalNAc-T2 regulation of endothelial lipase (LIPG) might be relevant outside the skin niche. Several studies have shown that LIPG acts as a primary regulator of high-density lipoprotein cholesterol (HDL-C) levels in blood[53,54]. More recently it has been shown that blocking of the enzyme with a monoclonal antibody increases HDL-C levels in non-human primates in a phase 1 trial[55]. At the same time, several genome-wide association studies have found a link between *GALNT2* and blood cholesterol levels[56,57], and it has been shown that a loss of *GALNT2* reduces HDL-C levels in humans, non-human primates and rodents[16]. GalNAc-T2 sites on LIPG described here, have the potential being negative regulators of enzyme activity. This potential regulation could be caused by direct interactions with the catalytic domain, interference of proposed head-to-tail homodimer formation, or electrostatic repulsion between sialylated O-glycans and cell-surface proteoglycan anchors.

As site-specific regulation can only be assessed based on the quantification of single-site glycopeptides, we mainly focused on this subset of our data. However, this analysis does not cover the large follow-up effect some enzymes may have[58,59]. Quantification of multi-site glycopeptides can only be performed at the peptide level (and not the site-specific level), which showed that e.g. the close range follow-up enzyme GalNAc-T7 indeed has more multi-site hits as compared to the single-site data. A further exploration of the denser glycosylated regions is warranted, e.g. exploiting glycan-specific proteases in the study design[60].

In a select set of identified specific GalNAc-T targets we were able to estimate the occupancy of the O-glycosylation site in a native background. Determining absolute glycan site occupancy is arguably one of the most desired missing pieces in large scale O-glycoproteomic analyses. While, over the past decade, thousands of O-glycosylation sites in the human proteome have been reported and more and more is known about their regulation, there is very limited knowledge on the actual abundance of these glycosylation sites in the total pool of proteoforms. Although a low abundance of a specifically glycosylated proteoform does not directly imply a low functional relevance, the high relative abundance of such a glycoform has a fair chance of having biological impact when in addition regulated by a specific GalNAc-T. The challenge in determining site occupancy in a complex background is that no glycan-specific enrichment can be performed to decomplexify the material as this will distort the ratio between the glycosylated site and its non-glycosylated counterpart. Here, we demonstrated that it is achievable to estimate O-glycosylation site occupancies in the context of a complex proteomic sample when genetically limiting the possible O-glycan structures to single GalNAc residues. Additionally, single HexNAc peptide modifications are expected to have a very limited effect on the ionization efficiency of a peptide, largely excluding the possible underestimation of site occupancy due to instrumental limitations[34]. This approach revealed huge differences in site occupancies across the investigated set of O-glycosylated proteins and suggests an intricate regulation of O-glycan initiation, also at sites not dependent on any particular GalNAc-T. Importantly, we have determined consistent occupancies of identical regions on proteins of wild type and *COSMC* knockout origin, suggesting firm control at the initiation step regardless of the local glycomic context. Having the information of occupancy for O-glycosylation sites of interest in relation to different structural and functional motifs can help streamline the list of candidates for probing biological functions of site-specific glycans.

In conclusion, we combined the systematic mapping of GalNAc-T specificities in a human, tissue-forming, keratinocyte cell line with the assessment of individual site occupancies to identify leads for future studies discovering functions of site-specific O-GalNAc glycans in tissue differentiation and homeostasis. Our current findings will provide more insights in the indispensable roles of O-glycosylation in human (patho)physiology, which might play a role in the development of treatment strategies for a variety of diseases.

## Methods
### Cell lines and culture
N/TERT-1 immortalized human keratinocytes (male) were kindly provided by James G. Rheinwald's lab, Harvard Institute of Medicine, Brigham & Women's Hospital. Cells were maintained in K-SFM (Gibco, USA) supplemented with 25 µg/ml BPE (Gibco), 0.2 ng/ml EGF (Thermo Scientific), and 0.3 mmol/l CaCl$_2$ (Sigma) at 37 °C with 5% CO$_2$ as previously described[61]. Cells were cultured until they reached ~30% confluence, at which time they were passaged by trypsinizing with TrypLE (Gibco). For experiments where cells were grown to complete confluence, medium was shifted to 1:1 vol/vol K-SFM/DF-K (DMEM/F12) supplemented with 25 µg/ml BPE, 0.2 ng/ml EGF, and 2 mM L-glutamine (Thermo Fisher). HEK293T (ATCC, Cat# CRL-3216) cells were cultured in DMEM (Gibco) containing 10% FBS (HyClone) at 37 °C with 5% CO$_2$ and passaged at ~90% confluence.

N/TERT-1 *GALNT* knockouts were generated using CRISPR/Cas9 technology by targeting particular gene exons by validated gRNAs[62] or gRNAs predicted by GPP[63] (gRNAs listed in Supplementary Data 1). gRNAs were cloned using oligos (TAGC, Denmark) into lentiCRISPR-v2-Puro plasmid backbone (Addgene #52961). Directional cloning and insertion of the gRNA duplex using BsmBI and T4 ligase into the LentiCRISPR-v2 plasmid backbone was done as described earlier[64]. All plasmids were propagated in One Shot™ Stbl3™ Chemically Competent *E. coli* cells (Thermo Fisher). Endonuclease free plasmid preparations

were made using Midi Prep kit (Thermo Fisher). For lentivirus production HEK293T cells were seeded at $1 \times 10^5$ cells/well density in a six-well plate and grown for 72 h until 80–90% confluence. For transfection, 200 μL OPTI-MEM (Gibco), 8 μL of 1 mg/mL PEI (Sigma), 0.8 μg LentiCRISPR-V2-gRNA plasmid, 0.6 μg pCMV-VSV-G plasmid (Addgene #8454), and 0.6 μg psPAX2 plasmid (Addgene #12260) were mixed and incubated for 10 min at RT before added to the adherent HEK293T cells. After 24 h the transfection medium was replaced by K-SFM for N/TERT-1 transduction. Medium containing viral particles was collected 48/72 h post-transfection, i.e. when virus had accumulated for 24 h after medium change, and filtered (0.45 μm pore size). Filtered virus-containing medium was mixed 1:1 with fresh complete K-SFM and 1:1000 polybrene (Sigma), and used to transduce N/TERT-1 cells overnight. Selection of knockout cell lines with 1 μg/ml puromycin (Gibco) started 48–96 h after transduction including biweekly cell passaging. Single clones were obtained by serial dilution in 96 well plates and knockout clones were identified by IDAA using ABI PRISM™ 3010 Genetic Analyzer (Thermo Fisher) and Sanger sequencing (GATC, Germany). Three clones were selected for each gene with out of frame indel formation. IDAA results were analyzed using Peak Scanner Software V1.0 (Thermo Fisher)[21]. N/TERT-1 with silent mutations provided phenotypic control cell lines.

## Organotypic cultures

Organotypic cultures were prepared as described in detail previously[19,65,66]. Briefly, human fibroblasts were suspended in acid-extracted Type I collagen (4 mg/ml) and allowed to polymerize over a 1-ml layer of acellular collagen in six-well culture inserts with 3-μm-pore polycarbonate filters (BD Biosciences NJ, USA). Gels were allowed to contract for 4–5 days before seeding with $3 \times 10^5$ N/TERT-1 keratinocytes in DMEM/F12 raft medium supplemented with 1.5% FCS (HyClone), 5 μg/ml insulin (Sigma), 0.1 nM cholera toxin (Sigma), 400 ng/ml hydrocortisone (Sigma), 0.02 nM triiodothyronine, and 0.18 mM adenine (Sigma). Inserts were raised to the air-liquid interface 4 days after cell seeding, and media was changed every second day for additional 10 days. At least three independent experiments at different time points with three different clones of each knockout cell line were conducted, resulting in organotypic cultures with similar morphologies.

## Immunofluorescence labeling and imaging

For immunofluorescence staining N/TERT-1 organotypic cultures were fixed for 24 h at 4 °C in 10% neutral buffered formalin and paraffin embedded. In all, 3–5 μm sections were used for hematoxylin-eosin staining or immunofluorescence labeling. Heat-induced antigen retrieval was performed in Citrate buffer (pH = 6.0) before applying primary antibodies against Keratin 10 (Dako (Agilent), Cat#M7002, dilution 1:100) and Involucrin (Thermo Fischer, Cat# PA1-37934, dilution 1:200), followed by secondary fluorescently labeled antibodies (Molecular Probes, AlexaFluorTM488 conjugated goat anti-mouse IgG (Cat# A-11029) and AlexaFluorTM594 conjugated goat anti-rabbit IgG (Cat# A-11012), dilution 1:500). After washing three times with PBS, sections were incubated in 0.1 μg/ml DAPI (Sigma) for 8 min at RT, washed again three times with PBS, and mounted with ProLong Gold Antifade Reagent (Thermo Fisher). Images were collected using a Leica fluorescence microscope.

## Preparation of differential proteomes and glycoproteomes

For differential glycoproteomic analysis, three biological replicates of the N/TERT-1 GALNT knockout clones and three technical replicates of wild-type N/TERT-1 cell lines were expanded in three 100 mm Petri dishes to ~90–100% monolayer confluency and cell pellets were harvested by scraping. Glycoproteomic sample preparation methods were adapted from previous reports[29]. Cell pellets were lysed by sonication in 0.5 mL 0.1% RapiGest SF surfactant (Waters, USA) in ammonium bicarbonate and inactivated by heating to 80 °C. Extracted proteins were reduced with dithiothreitol (Sigma), alkylated with iodoacetamide (Sigma) and digested with trypsin (1:40) (Roche, Switzerland) overnight at 37 °C. Digested samples were acidified using trifluoric acid (TFA) to quench the RapiGest SF and inactivate trypsin, and desalted on Sep-Pak C18 1cc 100 mg columns (Waters). Purified peptides were quantified using colorimetric peptide quantification kit (Thermo Fisher) and 400 μg peptide from each sample were labeled using 1.6 mg tandem mass tag (TMT) sixplex labels (Thermo Fisher). Labeled peptide samples were then mixed in equal amounts and treated with neuraminidase from Clostridium perfringens (Sigma) to remove sialic acids. O-glycopeptides (T and Tn) were enriched by lectin weak affinity chromatography (LWAC) using agarose-bound jacalin lectin (Vector Labs). After washing, bound glycopeptides were eluted with 0.8 M D-galactose. Eluted fractions were desalted using self-made Stage Tips (C18 and C8 sorbent from Empore 3 M), and fractionated using Pierce high pH reversed-phase peptide fractionation kit (Thermo Fisher). About 100 μg of LWAC flow-through were also fractionated and used for differential proteomics. Both glycoproteome and proteome samples were then submitted to LC-MS/MS.

## Preparation of O-glycopeptide occupancy samples

Three technical replicates of both wild type and COSMC knockout N/TERT-1 cells were grown in 100 mm Petri dishes. When cells reached ~50% confluency, media was exchanged with 7 ml K-SFM without BPE supplement. After 24 h, conditioned media was collected and new BPE-free media was added. This process was then repeated twice more, at which point cells would be close to ~100% confluency and a total of 20-21 ml conditioned media had been collected. Cells were washed once in 10 ml PBS, and cell pellets were collected using a cell scraper. Conditioned media was passed through a 0.45 μM filter membrane and proteins were precipitated by mixing with 100% (1 g/ml) trichloroacetic acid (TCA) in a 10:1 ratio (sample:TCA). Solutions were kept on ice for 1 h allowing proteins to precipitate before centrifugation at 21.000 × g at 4 °C for 15 min. Supernatants were discarded and protein pellets were washed twice with ice-cold acetone with centrifugation steps in between at 21.000 × g at 4 °C for 10 min. After removal of acetone from the second wash step the protein pellets were left to air dry. Pellets were resuspended in 0.05% RapiGest SF surfactant in 50 mM ammonium bicarbonate buffer and incubated at 70 °C for 20 min followed by reduction with 5 mM dithiothreitol and alkylation with 10 mM iodoacetamide. From each sample, 300 μg protein was digested with trypsin (Roche) overnight at 37 °C using a 1:40 trypsin to protein ratio. Total cell lysates (TCL) from cell pellets were prepared as described for differential glycoproteomes in the previous section. After digestion, both secretome samples and TCLs were desalted on Sep-Pak C18 1cc 100 mg columns (Waters). Eluted peptides were dried and reconstituted in 50 mM ammonium bicarbonate buffer followed by overnight incubation with 3 U PNGase F (Roche) at 37 °C. Peptides were desalted on Sep-Pak C18 and quantified using a colorimetric peptide quantification kit (Thermo Fisher). Before LC-MS/MS, 100 μg of each sample were separated into four fractions (12.5%, 17,5% 22.5% and 50% acetonitrile) using Pierce high pH reversed-phase peptide fractionation kit (Thermo Fisher).

## Preparation of differential glycosecretomes

Protein samples for differential glycosecretomes were prepared using TCA precipitation as described for occupancy samples. Two biological replicates were used for both N/TERT-1 GALNT1 and GALNT2 knockouts and two technical replicates were used for wild-type control samples. After precipitation, samples were treated similarly as for total cell lysate glycoproteomes. In short, samples were reconstituted in 0.05% RapiGest SF surfactant in 50 mM ammonium bicarbonate buffer and 400 μg secreted protein was reduced, alkylated, and digested with trypsin (1:40) overnight at 37 °C. After Sep-Pak C18 clean-up, 200 μg peptide was labeled using 0.8 mg TMT sixplex labels and mixed in

equal amounts. The combined sample was treated with neuraminidase prior to jacalin LWAC enrichment of glycopeptides. Both eluted glycopeptides and flow-through material were desalted using Stage Tips before finally being high pH fractionated and submitted to LC-MS/MS.

## Mass spectrometry

TMT-labeled proteome and glycoproteome samples were analyzed as previously described[23]. Briefly, samples were analyzed using an EASY-nLC system (Thermo Fisher Scientific) coupled to, either an Orbitrap Fusion Lumos Tribrid mass spectrometer (Thermo Fisher Scientific) via a nanoSpray Flex ion source (Thermo Fisher Scientific), or an Orbitrap Fusion mass spectrometer (Thermo Fisher Scientific) via a PicoView nanoSpray ion source (New Objectives). The nanoLC system was operated using a single analytical column setup packed with Reprosil-Pure-AQ C18 phase (Dr. Maisch, 1.9 μm particle size, 17–19 cm column length) in a PicoFrit Emitter (New Objectives, 75 μm inner diameter).

All samples, dissolved in Buffer A (0.1% formic acid, FA), were injected (12 μL for glycoproteome samples and 2 μL for proteome samples) onto the column. For the Lumos setup (nLC-1200), samples were eluted in a gradient from 3 to 32% Buffer B (80% acetonitrile, 0.1% FA) in 95 min, from 32 to 100% in 10 min, followed by isocratic elution at 100% for 15 min (total elution time 120 min). For the Fusion setup (nLC-1000), samples were eluted in a gradient from 2 to 20% Buffer B (100% acetonitrile, 0.1% FA) in 95 min, from 20 to 80% in 10 min, followed by isocratic elution at 100% for 15 min (total elution time 120 min). The mass spectrometers were set to acquire full scan MS spectra (*m/z* 355-1700; positive polarity) for a maximum injection time of 50 ms on Lumos, and 100 ms on Fusion, at a mass resolution of 120 K and an automated gain control (AGC) target value of $2.0–4.0 \times 10^5$. Higher-energy C-trap dissociation (HCD)-only and HCD + electron-transfer/collision-induced dissociation (ETciD) methods were used for proteome and glycoproteome samples, respectively, with a cycle of 10 scans in data-dependent mode. In HCD scans, the AGC target value was set to $5.0 \times 10^4$ and stepped collision energy was used with resolution at 50 K. In ETD scans, the AGC target was set to $1 \times 10^5$ and the maximum injection time was 150 ms with resolution at 60 K. All MS/MS spectra were acquired in the Orbitrap in profile mode. Charge states 2–6 on Lumos, and 2–7 on Fusion, were targeted for fragmentation, using dynamic exclusion with an exclusion window of 25 ppm.

For samples used for occupancy measurements, two microliter per sample (10% of total) was injected per analysis onto the Orbitrap Fusion Lumos Tribrid mass spectrometer-set up as described above. A 2 h LC method was used with a gradient from 3% to 32% of solvent B in 95 min, from 32% to 100% B in the next 10 min and 100% B for the last 15 min at 200 nL/min. A precursor MS scan (*m/z* 350–1700, positive polarity) was acquired in the Orbitrap at a nominal resolution of 120 K, followed by Orbitrap HCD-MS/MS at a nominal resolution of 50 K of the 15 most abundant precursors in the MS spectrum (charge states 2–6). A minimum MS signal threshold of 30,000 was used to trigger data-dependent fragmentation events. HCD was performed with an energy of 27% ± 5%, applying a 20-s dynamic exclusion window.

## Data analysis

MS data processing for all raw files was performed using Proteome Discoverer (PD) version 2.3 software (Thermo Fisher Scientific) and further data analysis was done in R with in-house scripts and Excel. For the differential glycoproteomics data, raw files were searched with Sequest HT search engine against a concatenated human-specific database (UniProt, March 2019, contacting 20,355 canonical entries). Enzyme restriction was set to trypsin digestion with full specificity, and a maximum of two missed cleavages. The precursor mass tolerance was set to 10 ppm and fragment ion mass tolerance to 0.02 Da. Carbamidomethylation on cysteine residues was used as a fixed modification. Methionine oxidation and HexNAc or Hex(1)HexNAc(1) attachment to serine, threonine, and tyrosine were used as variable

modifications, with a maximum of 10 variable modifications per peptide. The ptmRS node was used to determine the probability of glycosylation site localization. TMT6plex was set as a static modification on any peptide N-terminus and lysine residues. The Reporter Ion Quantifier node was applied for TMT6plex quantification of reporter ions using total peptide level normalization. Glycopeptide identification, and site localization was based on the ETD data, and ptmRS probability of > 95% was used as quality control cut-off. Reporter-ion glycopeptide quantification was based on the HCD spectrum of the same precursor as the highest quality ETD PSM per glycopeptide. To compare the relative abundance of glycopeptides between wild type and *GALNT* knockout samples, the fold change between the median of the replicates was calculated. Furthermore, a two-sided Student's *t* test was used as a measure of statistical confidence and the *p*-values were adjusted for multiple testing, using an FDR of 5%. For all 325 sites that were identified to be differentially regulated by one or more of the GalNAc-T enzymes, manual evaluation was performed of the ETD spectra to confirm the location of the glycan. For this we considered the ETD diagnostic c and z ions (ppm ± 1, min. 2 isotopologue peaks) proving both the presence of a glycan on a specific S, T, or Y residue, as well as excluding the possibilities of being present on any other S, T, or Y in the peptide (Supplementary Data 2). For the differential proteomics data, similar search and quantification settings were used as described above, except that the variable glycan modifications were excluded. To compare the relative abundance of proteins between wild type and *GALNT* knockout samples, the fold change between the median of the replicates was calculated.

Domain annotations were performed based on the information derived from UniProt, using the data from Prosite, which considers structural domains (i.e. characterized by their fold)[26].

For occupancy studies, *COSMC* knockout sample raw files were first searched with Byonic search engine against a concatenated human-specific database (UniProt, March 2019, contacting 20,355 canonical entries). Enzyme restriction was set to trypsin digestion with full specificity, and a maximum of two missed cleavages. The precursor mass tolerance was set to 10 ppm and fragment ion mass tolerance to 0.02 Da. Carbamidomethylation on cysteine residues was used as a fixed modification. Methionine oxidation, asparagine deamidation and HexNAc (max 5) attachment to serine and threonine were used as variable modifications. Subsequently, wild type sample raw files were processed against a smaller protein database, compiled of proteins from which glycopeptides had been identified in *COSMC* knockout samples with a Byonic score of 200 or higher. The top-17 O-glycan modifications previously identified in wild type N/TERT-1 cells[35] were included as variable modifications, with a maximum of 3 variable modifications per peptide. MS/MS spectra from all consistent identifications in replicates with a Byonic score of >200 were manually validated. Relative quantification of the non-glycosylated peptide and the different glycoforms was based on the MS1 peak areas. The complete list of glycoforms per peptide was imported into Skyline 21.1.0.146 (ProteoWizard), using the Proteomics Interface. Extracted ion chromatograms were generated per (glyco)peptide, including the isotopologues which had a relative abundance above 20. Chromatographic peaks were manually selected based on accurate mass (> −1.5 ppm, <1.5 ppm), isotopic dot product (idotp; >0.85) and proximity in retention time (within 5 min) to the other glycoforms of the same peptide portion. Finally, total area normalization was performed for the complete set of glycoforms per peptide (i.e. [sum of MS1 peak area of all glycoforms]/[sum of MS1 peak area of all glycoforms+MS1 peak area of non-glycosylated peptide]) to obtain an approximation of the glycan occupancy per peptide portion.

## Reporting summary

Further information on research design is available in the Nature Research Reporting Summary linked to this article.

## Data availability

All the MS data files are available via the ProteomeXchange Consortium[67] with the data set identifier PXD036791 (see Supplementary Data 7 for raw MS data overview). Source data are provided with this paper.

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

## Acknowledgements
We thank Louise Rosgaard Duus and Karin Uch Hansen, University of Copenhagen, for their expert help with the cell culture and sequencing. This project has received funding from the European Research Council (ERC) under the European Union's Horizon 2020 research and innovation programme (GlycoSkin H2020-ERC; 772735; M.I.N., N.d.H., I.B., S.D., and H.H.W.) the European Commission (Imgene H2020 and Remodel; M.I.N., N.d.H., I.B., S.D., and H.H.W.), the Lundbeck Foundation (R313-2019-869; H.H.W.), the Danish National Research Foundation (DNRF107; S.Y.V. and H.H.W.), the National Science Foundation (Graduate Opportunities Worldwide Grant; W.K.), the Neye Foundation (H.H.W.), the Friis Foundation (H.H.W.), the Michelsen Foundation (H.H.W.), and the A.P. Møller og Hustru Chastine McKinney Møllers Fond til Almene Formaal (S.Y.V. and H.H.W.).

## Author contributions
M.I.N., N.d.H., W.K., S.D., and M.L. designed and performed the experiments. M.I.N., N.d.H., W.K., and Z.Y. carried out data analysis. M.I.N., N.d.H., I.B., and H.H.W. interpreted the data. M.C.J., I.B., S.Y.V., and H.H.W. supervised the work. M.I.N., N.d.H., and H.H.W. wrote the manuscript. I.B. and H.H.W. reviewed and edited the manuscript. All authors reviewed and approved the final work.

## Competing interests
The authors declare the following competing financial interests: H.H.W. owns stocks and is a consultant for and co-founder of EbuMab, ApS, Hemab, ApS, and GO-Therapeutics, Inc., all not involved in, or related to, the research performed in this study. All other authors declare no conflicts of interest.
