## [Peer Review File · Nature Communications]

REVIEWER COMMENTS

Reviewer #1 (Remarks to the Author):

In the manuscript by Neilsen et al, the authors investigate O-glycosylation in a tissue-forming human cell line N/TERT-1. Individual knockouts of various GalNAc-Ts were performed and it was demonstrated that these knockouts led to more pronounced phenotypes when compared to knockouts of enzymes related to elongation of O-glycosylation. Additionally, the authors mapped the O-glycosylation in these cell lines, demonstrating that the modification is found on many defined, unstructured, and mucin-like domains. Finally, the authors attempted to map the glycosylation residency of O-glycans. Overall, this is an incredibly important and understudied field, and this manuscript puts forth very interesting and useful findings. The experimental data put forth is fairly strong and the manuscript is well written.

Major concerns:

- The majority of this paper focuses on site-specific O-glycosylation, and it is largely agreed upon within the field of glycoproteomics that manual validation is necessary to ensure proper localization of O-glycosylation. The authors refer to “unambiguous” localization but do not mention the manual validation of their reported sites. This detracts significantly from the work, given that so many of the sites may be mis-assigned, and could dramatically affect the conclusions that the authors have made. The authors should manually validate their data and/or provide sufficient justification as to why that is unnecessary.
- Given that so many of the O-glycosites were found within mucin domains, it would be interesting for the authors to repeat the analysis while incorporating glycoproteases. This would presumably increase the number of glycopeptides that they can assign, especially within the mucin-like domains.
- Regarding the authors’ domain assignments, how many of the “other” (i.e., Fig 3B) protein domains were previously described as mucin domains? Additionally, why is “other” left out of Fig. 3C? It would be interesting to know how many glycosites were assigned on these domains.
- Similarly, in Figure 4, how many of these proteins were previously assigned as mucin-like proteins? Did the authors confirm previous findings and/or add to the list of mucin proteins?
- In Figure 5, the authors specifically analyze peptides with a single glycosite, and suggest that GalNAc-T1 and T2 show a high level of glycopeptide change, whereas GalNAc-T7 and T10 do not. Given that T1 and T2 are “early” GalNAcTs and are much more likely to generate singly modified glycopeptides, this would follow that their knockouts will generate more peptides that will change significantly. On the other hand, T7 and T10 are “late” GalNAcTs and thus will likely generate fewer singly modified glycopeptides. This is not discussed and should be addressed. Is there a reason why the authors focused solely on singly modified peptides? Were there glycopeptides found with multiple sites that changed in the T7 and T10 datasets?

- The authors have previously mapped GalNAcT KO cell lines, most notably in 10.1074/mcp.RA118.001121 ; thus, the novelty of this approach is somewhat lacking.

- It is well established that glycopeptides can be suppressed in the ionization process and that unmodified peptides ionize better than glycopeptides. This is an important point to address, given that the authors are directly comparing unmodified peptides to glycopeptides in order to determine the glycan site occupancy. It would be prudent to demonstrate that a single GalNAc does not affect ionization sufficiently, or at the very least mention that this is a potential drawback in their approach.

Minor concerns:

- In Figure 2A, what does the dashed line represent? I do not believe this was addressed in the text or the figure legend.

- In Figure 2B, does the yellow coloration reflect weak keratin staining? Also related to this figure, it is a surprising finding that loss of core 1 or 2 structures has less of an effect than individual GalNAcTs. Can the authors provide more speculation as to why they think this might be the case? I believe this warrants more discussion.

- The authors used charge states 2-6 and 2-7 in different mass spec runs. Please discuss the reasoning for the variation in methods.

- In Figure 6C, it is a bit difficult to see the glycan ratio. It would be easier to visualize if the authors made the protein cartoons smaller and the glycans bigger and/or if they changed the shading of the glycans to make the coloring more dramatic/obvious.

Reviewer #2 (Remarks to the Author):

In this manuscript, the authors generated 9 individual GALNT knockouts in human N/TERT-1 cell line and applied glycopeptide enrichment strategy to map the O-glycosylation sites regulated by individual GALNTs. Additionally, to evaluate the importance of O-glycosylation sites, the authors investigated the O-glycan occupancy in N/TERT-1 proteome under the native conditions, without enrichment of glycopeptides.

Major comments:

1. To study the O-glycosylation sites regulated by individual GALNTs, the authors used jacalin LWAC to enrich glycopeptides. In GALNT KO samples, the glycoproteins with greatly decreased glycosylation sites or eliminated glycosylation sites couldn't be enriched in this experiment. Thus,

the method may lead to the loss of important candidates, which are highly and specifically O-glycosylated by individual GALNTs.

2. To study the O-glycosylation sites regulated by individual GALNTs, the authors used the cell lysates from wild type and GALNT KO cells, but not included the secreted proteins. This may lose the important candidates glycosylated by GALNTs and cause the difficulties to understand the phenotypes caused by GALNT KO.

3. In the results, the authors showed MS data about O-glycosylation sites regulated by individual GALNTs. However, it seems that there are less connections between the phenotypes and the MS results. Is this caused by lack of secreted protein analysis and the lectin enrichment method? Can authors give some discussions? For example:

For GALNT1 and GALNT2 KO, the authors found that “lack of GalNAc-T1 and GalNAc-T2 only caused minor changes to the cellular morphology” compared with other GALNT KO. However, from MS results, the authors found “in the knockouts of GalNAc-T1, -T2, -T3 and -T6, we found the highest numbers of downregulated glycosites”. Additionally, the authors found “the knockout of GalNAc-T2 resulted in the most dramatic effect on the O-glycoproteome”.

Additionally, the authors found that ablation of GALNT3, 7, 11 resulted in similar phenotypes but there were largely non-overlapping glycosylation sites between these enzymes.

The authors found that the ablation of GALNT6 caused differentiation defects in N/TERT-1 cells, which corresponds to the previous results that GALNT6 is essential to prevent differentiation in colon cancer cell line, LS174T. However, the authors “did not find any overlap between GalNAc-T6-specific targets found in this study and previously published GALNT6 knockout data”.

The authors found that “the most pronounced effect observed in GalNAc-T18 knockout tissue”, but a limited number of significantly altered glycosylation sites were found in T18 KO cells.

Previous studies showed that T18 enhances activities of T2 and T10. Is there any overlapping of substrates between these enzymes in this study?

4. In Fig.2B and Fig.S1, the phenotypes of 3 different clones of GALNT KO were shown. However, for some GALNTs, 3 clones seemed to show different phenotypes in Fig.S1, such as GALNT1, GALNT2, GALNT7 and GALNT11. Can authors give some discussions? For MS assay, the authors used 3 different clones of individual GALNTs. Were the MS results of different clones same?

Did GALNT14 KO also generate a thick stratum corneum as T18 KO?

Are there any proliferation defects in GALNT KO cells?

5. The authors studied O-glycosylation site occupancy under the native condition and showed there was a large variation in O-glycan occupancy, including the sites specifically regulated by individual GALNTs. Does this result suggest that the decreased O-glycosylation sites identified from GALNT KO cells using enrichment methods may be caused by variable occupancy, not by GALNT KO?

6. In O-glycan occupancy study, is there difference in site occupancy between secreted proteins and the proteins from cell lysates? Are the sites from secreted proteins show higher occupancy?

Minor comments:

1. In Fig.3E, should the gray bar graph be "Other domain types" as shown in Fig.3B, not "All domain types"?

2. In Fig.5A legend, "438 sites (brown circle) were..." should be "329 sites" as shown in Fig.5A and main text.

3. On page 8, in line 197, "the individual GalNAc-Ts to the N/TERT-1 O-glycoproteome (Figure 1B)" should be "(Figure 1A)".

Point-by-point query-response/action list to the Reviewers comments

Reviewer #1

In the manuscript by Nielsen et al, the authors investigate O-glycosylation in a tissue-forming human cell line N/TERT-1. Individual knockouts of various GalNAc-Ts were performed and it was demonstrated that these knockouts led to more pronounced phenotypes when compared to knockouts of enzymes related to elongation of O-glycosylation. Additionally, the authors mapped the O-glycosylation in these cell lines, demonstrating that the modification is found on many defined, unstructured, and mucin-like domains. Finally, the authors attempted to map the glycosylation residency of O-glycans. Overall, this is an incredibly important and understudied field, and this manuscript puts forth very interesting and useful findings. The experimental data put forth is fairly strong and the manuscript is well written.

Query #1: The majority of this paper focuses on site-specific O-glycosylation, and it is largely agreed upon within the field of glycoproteomics that manual validation is necessary to ensure proper localization of O-glycosylation. The authors refer to “unambiguous” localization but do not mention the manual validation of their reported sites. This detracts significantly from the work, given that so many of the sites may be mis-assigned, and could dramatically affect the conclusions that the authors have made. The authors should manually validate their data and/or provide sufficient justification as to why that is unnecessary.

Response #1: Thank you for raising this concern, we agree with the reviewer that the manual validation of glycopeptides and annotated sites is crucial in glycoproteomics research. Initially, we decided to rely on a very strict automated validation of the ETD-based site annotation using the ptmRS node in Proteome Discoverer with a cut-off of 95%. With this approach we expect a close-to-absent annotation of false positives, on the cost of missing some falsely negative annotated glycopeptides (Taus, J. Proteome Res., 2011 & Yang, Mol. Syst. Biol., 2018). In the view of our very large dataset, with an abundance of glycopeptide identifications, we considered this a fair approach. However, because we recognize the utmost importance of having correctly annotated sites to draw conclusions on differentially regulated sites (and the dramatic effect of miss-assignments in this case) we now performed a manual evaluation (by three of the expert authors) of all 329 sites that we reported to be differentially regulated by one or more of the GalNAc-Ts. In this evaluation we considered the ETD diagnostic c and z ions proving both the presence of a glycan on

a specific S, T or Y, as well as excluding the possibilities of being present on other S, T or Y residues in the peptide. This resulted in the conclusion that five of the annotated sites were incorrect, and 32 were ambiguously assigned. With 91.5%, the vast majority of sites was correctly/unambiguously assigned.

Action #1: We included our findings on the manual validation in *Supporting Table S2*, detailing the data file (publicly available in the ProteomeXchange Pride repository) and scan number on which the annotation was based, the peptide length and modified amino acid, and the diagnostic ions on which we based our conclusion. All diagnostic ions were found with an absolute ppm error below one and a minimum of two isotopologue peaks. This approach was briefly described in the method section (page 24, line 683): *“For all 325 sites that were identified to be differentially regulated by one or more of the GalNAc-T enzymes, manual evaluation was performed of the ETD spectra to confirm the location of the glycan (unambiguous assignment). For this we considered the ETD diagnostic c and z ions (ppm \pm 1, min. 2 isotopologue peaks) proving both the presence of a glycan on a specific S, T or Y residue, as well as excluding the possibilities of being present on any other S, T or Y in the peptide (Table S2)”*.

Furthermore, the incorrectly assigned sites were removed from the results (and all numbers in the text were corrected accordingly), while the ambiguity of others were indicated in the text (page 8, line 201): *“Of these, we found 325 glycosylated sites (11.4% of total), of which 92% was unambiguously assigned to a specific amino acid in the peptide, with a significantly lower abundance in at least one of the GALNT knockouts, as compared to the wild type cells (fold change < 0.5 and False Discovery Rate (FDR) adj. p-value < 0.05).”*

Query #2: Given that so many of the O-glycosites were found within mucin domains, it would be interesting for the authors to repeat the analysis while incorporating glycoproteases. This would presumably increase the number of glycopeptides that they can assign, especially within the mucin-like domains.

Response #2: The use of glycoproteases is indeed a promising alternative and synergetic approach for the identification of O-glycosylation sites. While an important part of our work is focused on O-glycosylation site mapping, our main emphasis is on the differential regulation of these sites by the different GalNAc-Ts and their occupancy in a native situation. For the latter, we considered the use of glycoproteases less suitable, as non-glycosylated sites would be located on different peptide cleavage products as compared to occupied sites. As the description of the N/TERT-1 mucin-like

domains was not the initial aim of our study we did not include glycoproteases in the current study design. With that said, we agree that it would be highly interesting to investigate the mucinomes of different cells more in-depth in a future and separate study.

Action #2: We extended the discussion on the use of glycoproteases to study the mucinome at page 15, line 414: *“To further investigate the N/TERT-1 mucinome and its differential regulation by GalNAc-Ts, alternative protease cleavage regimes based on O-glycoproteases could be applied, which will likely increase the annotation of sites within mucin-like domains even further (Malaker, PNAS, 2019).”*

Query #3: Regarding the authors’ domain assignments, how many of the “other” (i.e., Fig 3B) protein domains were previously described as mucin domains? Additionally, why is “other” left out of Fig. 3C? It would be interesting to know how many glycosites were assigned on these domains. - Similarly, in Figure 4, how many of these proteins were previously assigned as mucin-like proteins? Did the authors confirm previous findings and/or add to the list of mucin proteins?

Response #3: We performed the domain annotation based on the UniProt-incorporated data from Prosite, which considers structural domains (i.e. characterized by their fold). Mucin “domains” are not considered as domains in this regard and are therefore not included in this analysis. Information on mucin-like domains in UniProt is limited, although sometimes indicated as a mucin “region” or Pro/Ser/Thr-rich region. While, to our knowledge, no extensive overview of mucin-like proteins exists (and the definition of “mucin domain” is somewhat scattered), we did compare our results to a list derived from recent literature (Malaker *et al.* 2022 Nat. Comm.) thoroughly investigating the human mucinome. This is mentioned in the following (page 15, line 403): *“The highly occupied mucin patches we identified on CD44, FN1, APP and DAG1 were predicted to be mucin domains previously. We also found potential mucin domains on other proteins, including FGFP1, EFNB1 and HSPA5.”* In addition to this, we now also included the comparison of our data to the UniProt-assigned mucin regions.

Action #3: We included a more accurate description of the domain annotation in the method section (page 24, line 692): *“Domain annotations were performed based on the information derived from UniProt, using the data from Prosite, which considers structural domains (i.e. characterized by their fold).”*

Further, we added the “other” domains to Figure 3C, showing that the majority of these domains also only carry one O-glycan site. Finally, we compared our list of proteins with mucin-like patches, in addition to recent literature (Malaker, Nat. Comm., 2022), to the proteins that are UniProt annotated to contain a mucin region or Pro/Thr/Ser-rich sequences. This comparison, including a better overview of identified glycan patches, has now been included in a new supplementary data table (Table S3). The comparison was included in the discussion (page 15, line 401): “*Out of the 68 proteins for which we identify glycan patches, 15 were described previously as having mucin-like domains (Malaker, Nat. Comm., 2022 & Bateman, Nucleic Acids Res., 2015) leaving 53 potential new mucin domain identifications (Table S3).*”

Query #4: In Figure 5, the authors specifically analyze peptides with a single glycosite, and suggest that GalNAc-T1 and T2 show a high level of glycopeptide change, whereas GalNAc-T7 and T10 do not. Given that T1 and T2 are “early” GalNAcTs and are much more likely to generate singly modified glycopeptides, this would follow that their knockouts will generate more peptides that will change significantly. On the other hand, T7 and T10 are “late” GalNAcTs and thus will likely generate fewer singly modified glycopeptides. This is not discussed and should be addressed. Is there a reason why the authors focused solely on singly modified peptides? Were there glycopeptides found with multiple sites that changed in the T7 and T10 datasets?

Response #4: Thank you for this insightful comment. We now included a more extensive discussion on this topic by highlighting the supplementary data describing the analysis of the multi-glycosylated peptides (Supplementary figure S3 and Supplementary table S2).

The reason for our focus on single-glycosites is that our differential approach inherently hampers the quantification of sites on multiple-glycosylated peptides. The decrease in abundance of a multi-modified peptide cannot be directly connected to a specific site, and it is therefore only possible to study multi-glycosylated sites at the peptide level (and not at the site-specific level). Our analysis of this data shows that T7 and T10 indeed have more multi-site hits as compared to the single-site data. To specifically investigate which sites are affected on these peptides, single-case evaluation is required, and we considered this outside the scope of the current, already rather comprehensive, study. As a note, all sites, including the ones on multi-glycosite peptides, were included in the domain and mucin-patch annotations.

Action #4: We improved the description of the multi-site data in the results (page 10, line 275): “*This can be partly explained by some of these enzymes (GalNAc-T7 and -T10) being “late”*”

GalNAc-Ts, responsible for follow-up glycosylation of previously glycosylated regions. This was indeed reflected in the analysis of the multi-glycosylated peptides in our data (Figure S3, Table S2), where e.g. additional T7- regulated regions were found for the proteins FGFBP1, AGRN, ERP44 and COL17A1.”

Additionally, we extended the discussion on this data (page 17, line 475): “As site-specific regulation can only be assessed based on the quantification of single-site glycopeptides, we mainly focused on this subset of our data. However, this analysis does not cover the large “follow-up” effect some enzymes may have (Raman, JBC, 2008 & Revoredo, Glycobiology 2016).

Quantification of multi-site glycopeptides can only be performed at the peptide level (and not the site-specific level), which showed that e.g. the close range follow up enzyme GalNAc-T7, indeed has more multi-site hits as compared to the single-site data. A further exploration of the denser glycosylated regions is warranted, e.g. exploiting glycan-specific proteases in the study design (Shon, Biochem. J., 2021).”

Query #5: The authors have previously mapped GalNAcT KO cell lines, most notably in 10.1074/mcp.RA118.001121 ; thus, the novelty of this approach is somewhat lacking.

Response #5: The approach for mapping GalNAc-T-specific regulation of glycosylation sites is indeed well established. We here aimed to apply this recognized approach in combination with other approaches to go beyond the current knowledge on O-glycan site location, regulation and occupancy. What, in our opinion, makes our study novel is: 1) The application of the differential method on a tissue-forming keratinocyte model with relevance for true human biology, including the phenotypic readouts from the knockout tissues in Figure 1 and Figure S1. 2) The inclusion of the complete panel of expressed GalNAc-Ts in our knockout approach, allowing the better definition of transferase specificities. 3) The development of an approach to assess the occupancy of glycosylation sites, using non-enriched material, which – at least for some sites – allows us to combine site-specific regulation and occupancy for the identification of putative mechanistically relevant sites. 4) The detailed description of mucin-like patches in the glycoproteome. 5) The fact that our data is highly systematic and based on three biological knockout replicates per condition.

Query #6: It is well established that glycopeptides can be suppressed in the ionization process and that unmodified peptides ionize better than glycopeptides. This is an important point to address, given that the authors are directly comparing unmodified peptides to glycopeptides in order to

determine the glycan site occupancy. It would be prudent to demonstrate that a single GalNAc does not affect ionization sufficiently, or at the very least mention that this is a potential drawback in their approach.

Response #6: Thank you for highlighting this concern and we agree that this is an important point to address. We have now improved the discussion on our approach accordingly. As said, we agree that this is an important point in the field which should be treated with careful consideration, however, we would like to advocate that the effect of a single GalNAc on the ionization of a (glyco)peptide is limited. While it is well established that an N-glycan attached to a peptide reduces the ionization/desolvation efficiency of a peptide as compared to its non-glycosylated counterpart, it is shown that a single GlcNAc modification did not have this effect. This was true, independently from GlcNAc location in the peptide sequence and instrumentation used (Stavenhagen, J. Mass Spectrom, 2013). We believe these data can be extrapolated to the single HexNAc modification with a GalNAc. What strengthens us in this belief is that our occupancy data based on the *COSMC* KO material (displaying single GalNAc O-glycans) resulted in occupancy values that were highly similar to the ones we could assess in WT material (displaying branched and sialylated O-glycans); see Figure 6B. This supports that small glycans (< 7 monosaccharides) have limited effect on the ionization efficiency of the peptides, even when carrying sialic acids. However, the latter might be dependent on the peptide moiety and should be investigated further (something we considered out of scope for our study that mainly bases occupancy on *COSMC* KO material).

Action #6: In the text we adjusted the following in the results (page 11, line 293): “*COSMC is essential for elongation of O-glycans with galactose beyond the initial GalNAc-residue and its deletion simplifies the repertoire of O-glycans, as well as reduces the possible bias between the quantification of the peptides and their glycosylated variants* (Stavenhagen, J. Mass Spectrom, 2013).”

And in the discussion (page 18, line 496): “*Additionally, single HexNAc peptide modifications are expected to have a very limited effect on the ionization efficiency of a peptide, largely excluding the possible underestimation of site occupancy due to instrumental limitations* (Stavenhagen, J. Mass Spectrom. 2013).”

Query #7: In Figure 2A, what does the dashed line represent? I do not believe this was addressed in the text or the figure legend.

Response #7: Thank you for the comment.

Action #7: We removed the line and the color from the figure, as these were not required for proper interpretation of the figure.

Query #8: In Figure 2B, does the yellow coloration reflect weak keratin staining? Also related to this figure, it is a surprising finding that loss of core 1 or 2 structures has less of an effect than individual GalNAcTs. Can the authors provide more speculation as to why they think this might be the case? I believe this warrants more discussion.

Response #8: The yellow color indicates simultaneous staining of Keratin10 (green; staining all suprabasal cells) and Involucrin (red; staining the more differentiated cells). Regarding the finding that loss of core 1 or 2 structures has less of an effect than individual GalNAc-Ts, we speculate that this is induced by the fact that the loss of complete glycans, including the initiation step, will affect localization, surface expression and secretion of specific proteins, while loss of glycan elongation mainly affects cell-cell interactions between epithelial or with other cell types, such as immune cells and melanocytes not present in our system. The effect on cell-cell interaction between epithelial cells is supported by our previous findings that loss of O-glycan elongation beyond the initial GalNAc-residue affects cell-cell interaction, initial differentiation, and tissue fragmentation under mechanical stress (Dabelsteen, Dev. Cell, 2020), and in combination with cancer associated mutations results in an invasive phenotype (Radhakrishnan, PNAS, 2014).

Action #8: We extended the discussion by (page 13, line 351): *“Here, it needs to be noted that the model only reflects the endogenous effects of glycans in differentiation and tissue-formation of the keratinocytes, and not the interaction with other cell types such as immune cells. The loss of complete glycans most likely affect localization, surface expression and secretion of specific proteins, while loss of all elongated glycans and glycan decoration might play more significant roles in tissue stability under stressed conditions and in the interaction between different cell types (Dabelsteen, Dev. Cell, 2020).”*

Query #9: The authors used charge states 2-6 and 2-7 in different mass spec runs. Please discuss the reasoning for the variation in methods.

Response #9: The difference in charge states were a result of different method settings on our MS instruments (Orbitrap Fusion Lumos vs. Orbitrap Fusion) and were as such not intended. We did consider the possible effect this could have on the acquired data, and we found that 7+ charged species found in the Orbitrap Fusion runs accounted for less than 0.2% of the total identifications. Additionally, we did not find any precursors uniquely present in the 7+ charge state. We therefore do not believe that this difference in precursor selection has any significant effect on the data obtained.

Query #10: In Figure 6C, it is a bit difficult to see the glycan ratio. It would be easier to visualize if the authors made the protein cartoons smaller and the glycans bigger and/or if they changed the shading of the glycans to make the coloring more dramatic/obvious.

Response and action #10: We agree that the suggested changes would improve Figure 6C. We have therefore reduced the size of the protein cartoons, increased the size of the glycans, and changed the glycan color gradient to make the glycosylated ratios more obvious.

Reviewer #2

In this manuscript, the authors generated 9 individual GALNT knockouts in human N/TERT-1 cell line and applied glycopeptide enrichment strategy to map the O-glycosylation sites regulated by individual GALNTs. Additionally, to evaluate the importance of O-glycosylation sites, the authors investigated the O-glycan occupancy in N/TERT-1 proteome under the native conditions, without enrichment of glycopeptides.

Query #1: To study the O-glycosylation sites regulated by individual GALNTs, the authors used jacalin LWAC to enrich glycopeptides. In GALNT KO samples, the glycoproteins with greatly decreased glycosylation sites or eliminated glycosylation sites couldn't be enriched in this experiment. Thus, the method may lead to the loss of important candidates, which are highly and specifically O-glycosylated by individual GALNTs.

Response #1: Thank you for sharing your concern. We overcame the implied loss of important candidates by including isobaric Tandem-Mass-Tag labeling in our study design prior to the jacalin LWAC enrichment as described previously (Bagdonaite, EMBO Reports, 2020). Each sample contains peptides from 3 wildtype replicates and 3 *GALNT* knockout replicates, and in this way the wildtype samples act as a reference. The multiplexing prior to the enrichment, ensures that only one of the original samples is required to contain the glycopeptide of interest for it to be picked up and quantified in all six samples. In those cases where glycosylation is completely lost in the *GALNT* knockout, the jacalin LWAC will capture wildtype glycopeptides, and TMT-reporter ions will then only be obtained from wildtype reporter channels.

Query #2: To study the O-glycosylation sites regulated by individual GALNTs, the authors used the cell lysates from wild type and GALNT KO cells, but not included the secreted proteins. This may lose the important candidates glycosylated by GALNTs and cause the difficulties to understand the phenotypes caused by GALNT KO.

Response #2: This is a relevant point and we agree that we might miss important GalNAc-T targets by only focusing on proteins from total cell lysates. We choose this initial approach because it covers the highest number of glycoproteins from throughout the cells and it is also the most established sample type in the field. However, we have now optimized our approach for the analysis of secretome samples and included the differential analysis of the *GALNT1* and *-T2* knockout secretome for comparison with the lysate samples. This data gives a good idea of the similarities between the two sample types, which is largely explained by the high enrichment efficiency of our Jacalin LWAC approach. The new data is represented in a new supplementary Figure S3 and the data has also been added to Table S2. Overall, while some extra glycan site identification were made in the secretome, the data showed a large overlap in regulated glycan site identifications when compared to the total lysate samples.

Action #2: We extended our results by (page 10, line 259): “As *O*-glycosylation is an abundant modification of secreted proteins (Steentoft, EMBO J., 2013), we evaluated the overlap between our results in the total cell lysates and the differential glycoproteomics of secreted material from *GALNT1* and *GALNT2* knockouts (Figure S3 and Table S2). We found that the majority of glycoproteins (86%) and glycosites (77%) identified in the secretome samples were covered by the total cell lysate approach, which suggests that the enrichment strategy effectively picks up

glycopeptides from secreted glycoproteins in total cell lysate samples. The secretome added 32 new glycoprotein identifications (3.7% of total) and 133 new glycosylation site identifications (4.4% of total), of which 14 were regulated by either GalNAc-T1 or -T2. The knockout of GalNAc-T1 had a relatively small impact on the glycosome, while the impact of the knockout of GalNAc-T2 was substantial. Additional T2-specific sites found in the secretome included PXDN Thr¹³⁴⁸ in the region important for homotrimerization of the protein, and on the propeptide of MMP1 (Thr²³). Important T2-regulated sites found in the lysate samples, including PRCP (Thr³⁹), LIPG (Thr⁴¹), CTSD (Thr⁶⁷), and JAG1 (Thr⁹⁰⁴), were confirmed in the glycosome”

The materials & methods section has also been updated to include information on the differential glycosomes.

Query #3: In the results, the authors showed MS data about O-glycosylation sites regulated by individual GALNTs. However, it seems that there are less connections between the phenotypes and the MS results. Is this caused by lack of secreted protein analysis and the lectin enrichment method? Can authors give some discussions? For example:

For GALNT1 and GALNT2 KO, the authors found that “lack of GalNAc-T1 and GalNAc-T2 only caused minor changes to the cellular morphology” compared with other GALNT KO. However, from MS results, the authors found “in the knockouts of GalNAc-T1, -T2, -T3 and -T6, we found the highest numbers of downregulated glycosites”. Additionally, the authors found “the knockout of GalNAc-T2 resulted in the most dramatic effect on the O-glycoproteome”.

Additionally, the authors found that ablation of GALNT3, 7, 11 resulted in similar phenotypes but there were largely non-overlapping glycosylation sites between these enzymes.

The authors found that the ablation of GALNT6 caused differentiation defects in N/TERT-1 cells, which corresponds to the previous results that GALNT6 is essential to prevent differentiation in colon cancer cell line, LS174T. However, the authors “did not find any overlap between GalNAc-T6-specific targets found in this study and previously published GALNT6 knockout data”.

The authors found that “the most pronounced effect observed in GalNAc-T18 knockout tissue”, but a limited number of significantly altered glycosylation sites were found in T18 KO cells.

Previous studies showed that T18 enhances activities of T2 and T10. Is there any overlapping of substrates between these enzymes in this study?

Response #3: Thank you for pointing out these concerns. We would not expect to find a direct correlation between biological function, i.e., a phenotypic effect, and the number of identified O-glycosylation sites. Many examples demonstrate that single O-glycan sites have a pronounced effect due to important functions on a specific set of proteins; for example, the rather selective effect of the broadly expressed GalNAc-T3 on calcium regulation through the protection of FGF23 from furin cleavage. Another example is the similarly selective effect of GALNT6, but not the homologous GALNT3, on differentiation as observed in the colon cancer cell line, LS174T (Lavrsen, JBC, 2018); an effect that corresponds well to the differentiation phenotypes observed here for GALNT6 in human skin. Together with the observed impact on LS174T and other epithelial models it suggests that GALNT6 has an essential function in epithelial differentiation. Although this study does not reveal which of the identified GALNT6 targets are responsible, we are hopeful that future studies can help explain the molecular mechanisms behind the observed phenotype.

It is also true, as stated by the reviewer, that multiple knockouts lead to phenotypes with similar traits, for example as observed in models created with *GALNT3*, *T7*, and *T11* KO cells. However, it is well known that alterations in different molecular pathways may lead to the same gross phenotypes during the differentiation and transformation of human skin. For example, changes in either p53, TGF signaling pathway, and HRAS all cause invasion but, in each case, through different signaling pathways. Therefore, only a detailed study of the affected signaling pathways will allow an appreciation of the differences between the effects of GALNT3, T7, and T11 and the seemingly similar phenotypes. We are hopeful that, for example, differential phospho-proteomics in combination with the identification of distinct high occupancy substrates will allow us to dissect further the individual (signaling) pathways affected by each of the GALNTs.

Finally, regarding the potential overlap of substrates between T18 and T2/T10, this is an interesting suggestion. We indeed found 10 of the T2 and 2 out of the 3 T10 targets to overlap with the T18 targets. We included this in the discussion.

Action #3:

1) Included text (page 16, line 442): *“Based on the largely non-overlapping and rather discrete substrate specificities between the four enzymes, the phenotypic consequences most likely result from an effect on distinct pathways.”*

2) Included text (page 16, line 451): *“Indeed we found 10 of the GalNAc-T2 and 2 out of the 3 GalNAc-T10 targets to overlap with the T18 targets.”*

Query #4: In Fig.2B and Fig.S1, the phenotypes of 3 different clones of GALNT KO were shown. However, for some GALNTs, 3 clones seemed to show different phenotypes in Fig.S1, such as GALNT1, GALNT2, GALNT7 and GALNT11. Can authors give some discussions? For MS assay, the authors used 3 different clones of individual GALNTs. Were the MS results of different clones same? Did GALNT14 KO also generate a thick stratum corneum as T18 KO? Are there any proliferation defects in GALNT KO cells?

Response #4: Indeed, some differences were observed between the different clones of the knockouts, which would be expected from 3D models built with biological replicates. Based on the helpful comment, we realize we did not sufficiently control the relative amount of K10 (green) and Involucrin (red) expression, which also affects how the images are displayed. We have now re-sectioned and re-labeled all KO skin models again and taken new images from 3 representative areas from the models. We believe that the new figure now better represents the phenotype of the individual KO tissue models. Despite, some differences, the overall phenotypes of the individual clones are relatively consistent. It should be noted that the stratum corneum is often affected by the sectioning and preparation of the tissue and hence not considered in our phenotypic description. Finally, we do not see any apparent changes in proliferation among the different knockout cell lines, although this has not been assessed in detail.

Regarding the MS results of the different clones, we found that they were very similar and we only included results that were statistically significant when comparing the three clones together to the WT triplicates. This is described in the method section (page 24, line 680): *“To compare the relative abundance of glycopeptides between wild type and GALNT knockout samples, the fold change between the median of the replicates was calculated. Furthermore, a two-sided Student's t-test was used as a measure of statistical confidence and the p-values were adjusted for multiple testing, using an FDR of 5%.”*

Action #4: Figure S1 has been updated with new images and stainings which better represent the observed phenotypes.

Query #5: The authors studied O-glycosylation site occupancy under the native condition and showed there was a large variation in O-glycan occupancy, including the sites specifically regulated by individual GALNTs. Does this result suggest that the decreased O-glycosylation sites identified from GALNT KO cells using enrichment methods may be caused by variable occupancy, not by GALNT KO?

Response #5: Thank you for highlighting that this was unclear. We meant to write that comparing different sites to each other, different occupancies could be observed, ranging from highly occupied sites to very lowly occupied ones. On the other hand, the occupancy of the same site in different samples was very robust. The repeatability between the replicates was indicated in Table S4 (columns SDglyco and CVglyco) and by the errorbars in Figure 6A. This ensures us that the decreased O-glycosylation sites identified in the KO cells are a result of the KO and not of variation in the occupancy. Additionally, if such variation would be present, the replicates would have prevented the findings from being statistically significant.

Action #5: We clarified the results as follows (page 12, line 313): *“The complete set of identified pairs of glycosylated and non-glycosylated variants from different protein targets in COSMC knockout samples showed a large variation in O-glycan occupancies between different glycosylation sites on different proteins, while replicates of the same site were consistent.”*

Query #6: In O-glycan occupancy study, is there difference in site occupancy between secreted proteins and the proteins from cell lysates? Are the sites from secreted proteins show higher occupancy?

Response #6: This is a very interesting question. Unfortunately, due to technical limitations, we were unable to assess enough targets in the total cell lysate to make a proper comparison to the secretome. This was the main reason that we shifted to the secretome analysis for site occupancy determination in the first place. The secretome has a natural enrichment of glycoproteins, which allows the assessment of occupancy in the otherwise too complex background.

Query #7: In Fig.3E, should the gray bar graph be “Other domain types” as shown in Fig.3B, not “All domain types”?

Response #7: Thank you for checking. In 3B “other” shows number of sites other than those highlighted. In 3E “All domain types” is in fact all domain types – so including those shown below.

Action #7: We changed the color of the graph to prevent confusion.

Query #8: In Fig.5A legend, “438 sites (brown circle) were...” should be “329 sites” as shown in Fig.5A and main text.

Response #8: Thank you for noting this mistake.

Action #8: We corrected the number in the legend.

Query #9: On page 8, in line 197, “the individual GalNAc-Ts to the N/TERT-1 O-glycoproteome (Figure 1B)” should be “(Figure 1A)”.

Response #9: Thank you for noting this mistake.

Action #9: We corrected the figure reference.

REVIEWERS' COMMENTS

Reviewer #1 (Remarks to the Author):

The authors have very carefully and thoroughly addressed all of my previous concerns and suggestions. I greatly appreciate their manual validation of glycosites, inclusion of Table S3, and the additional glycosite data.

Reviewer #2 (Remarks to the Author):

In the revisions, the authors addressed most reviewers concerns. The revised manuscript looks good.